# An endoribonuclease-based feedforward controller for decoupling resource-limited genetic modules in mammalian cells

Ross D. Jones [1,2], Yili Qian [2,3], Velia Siciliano [1,2,5], Breanna DiAndreth [1,2], Jin Huh[1,2], Ron Weiss [1,2,4✉] & Domitilla Del Vecchio [2,3✉]

Synthetic biology has the potential to bring forth advanced genetic devices for applications in healthcare and biotechnology. However, accurately predicting the behavior of engineered genetic devices remains difficult due to lack of modularity, wherein a device's output does not depend only on its intended inputs but also on its context. One contributor to lack of modularity is loading of transcriptional and translational resources, which can induce coupling among otherwise independently-regulated genes. Here, we quantify the effects of resource loading in engineered mammalian genetic systems and develop an endoribonuclease-based feedforward controller that can adapt the expression level of a gene of interest to significant resource loading in mammalian cells. Near-perfect adaptation to resource loads is facilitated by high production and catalytic rates of the endoribonuclease. Our design is portable across cell lines and enables predictable tuning of controller function. Ultimately, our controller is a general-purpose device for predictable, robust, and context-independent control of gene expression.

[1] Department of Biological Engineering, Massachusetts Institute of Technology, Cambridge, MA 02139, USA. [2] Synthetic Biology Center, Massachusetts Institute of Technology, Cambridge, MA 02139, USA. [3] Department of Mechanical Engineering, Massachusetts Institute of Technology, Cambridge, MA 02139, USA. [4] Department of Electrical Engineering and Computer Science, Massachusetts Institute of Technology, Cambridge, MA 02139, USA. [5] Present address: Instituto Italiano di Tecnologia, Napoli 80125, Italy. ✉email: rweiss@mit.edu; ddv@mit.edu

A promising strategy for engineering complex genetic devices is to compose together simpler systems that have been characterized in isolation[1–4]. A critical assumption of this modular design approach is that subsystems maintain their input/output (i/o) behavior when assembled into larger systems. However, this assumption often fails due to context dependence, i.e., the behavior of a module depends on the surrounding systems[2,5]. There are many sources of context dependence, including unexpected off-target interactions between regulators and their targets[6–8], transcription factor (TF) loading by DNA targets[9,10], gene orientation[11], and resource loading by expressed genes[12,13]. To date, much effort has gone into identifying and engineering gene regulators with unique binding specificity, e.g., between TFs and their DNA-binding sites, with the goal of finding gene regulators that work orthogonally[7,14–18]. Nevertheless, even if subsystems are entirely composed of putatively orthogonal regulators, their gene expression levels can still become coupled to each other via competition for shared cellular resources[2,12,13,19,20]. For example, it has been demonstrated in prokaryotes that genes compete for ribosomes, such that increased expression from one gene decreases expression from others by sequestering, i.e., loading, ribosomes[12,13]. In mammalian cells, several types of cellular resources not present in prokaryotes are shared among expressed genes and can be overloaded, including transcription coactivator proteins (CoAs) and general TFs (GTFs) needed for transcription[21], splicing factors[22], miRNA-processing factors[23], RISC complexes[24,25], and the proteasome[26].

In particular, eukaryotic transcriptional activators (TAs) are known to apply a load to transcriptional resources by sequestering CoAs and/or GTFs from other genes, a phenomenon referred to as squelching[27–35]. This resource loading leads to a drop in the expression level of other genes, resulting in coupling between independently expressed genes and more generally to context-dependent gene expression. Moreover, squelching can be toxic to cells[34,36–38] and places a selective pressure against engineered circuits and the host cell, thus affecting both on evolutionary timelines[39,40]. As many established synthetic eukaryotic gene-regulation systems utilize TAs[14,17,41–44], squelching is potentially a pervasive problem in eukaryotic synthetic biology. Thus, we focus on characterizing the effects of resource loading by TAs and develop an engineering solution to make the expression level of a gene of interest (GOI) robust to resource loading.

We first establish an experimental model system to comprehensively quantify the effects of resource loading by different TAs on various human- and viral-derived constitutive promoters driving a GOI in different cell lines. From this characterization, we find that resource loading by the TAs substantially affects expression levels of the GOI in nearly all combinations of promoters, TAs, and cell lines tested. To provide a mechanistic understanding of the trends observed in the data, we build a mathematical model of eukaryotic gene expression which accounts for resource loading, including squelching by TAs. To solve the resource loading problem in mammalian cells, we introduce a feedforward controller design based on enzymatic regulation of the GOI to make its expression level robust to resource loading (Fig. 1a–d). Through a mechanistic model, we elucidate that the controller's ability to rescue the expression of the GOI back to the unperturbed level relies on fast catalytic and production rates of the regulating enzyme. Based on these design requirements, we chose the Cas6-family endoribonuclease (endoRNase) CasE[45,46] (EcoCas6e), as the regulating enzyme. In our design, CasE cleaves a 20 nt target site in the 5′ untranslated region (UTR) of the output mRNA, preventing translation. In a number of different cell lines and in response to resource loading

by a variety of TAs, our controller can maintain the desired expression level of the GOI, thereby demonstrating near-perfect adaptation of ectopic gene expression levels to resource loading in mammalian cells. Our controller thus represents a significant step toward engineering genetic systems in mammalian cells that function reliably regardless of their cellular context.

## Results

**Characterization of transcriptional resource sharing.** We first quantified the effect of resource sharing on the output levels of genetic devices. Specifically, we define a genetic device as an engineered gene that may take regulatory inputs (e.g., sequence-specific TFs) and gives the gene's expressed protein as output. We further define a genetic module as one or more genetic devices that are linked together by direct regulatory interactions. Independently-regulated devices in separate modules can become implicitly coupled through competition for shared gene expression resources: expression of a gene in one device "loads" the pool of shared resources, thereby decreasing resource availability to other devices in all modules (Fig. 1a). Because of this coupling, the behavior of a genetic device or module becomes dependent on the presence of devices in other modules in the cell.

We recapitulated resource sharing in mammalian cells using the genetic model system shown in Fig. 1e. The Gal4 DNA-binding domain (DBD) was fused to one of several activation domains (ADs) of varying potency (Supplementary Fig. 1), the strongest five of which were chosen for in-depth study: HSV-1 VP16[47], VP64[48], NF-$\kappa$B p65[49], EBV Rta[50], and the tripartite VP64-p65-Rta (VPR[51]). Our model system comprises two genetic modules (Fig. 1e). Module 1 comprises a device for constitutive expression (CMV:output$_1$). Module 2 comprises two devices: Gal4 TA expression (hEF1a:Gal4-{AD}) and Gal4-driven activation: UAS:output$_2$. The output and marker proteins are fluorescent reporters that we measured by flow cytometry. Typically, transfection markers (TX markers) are used for normalization of signals measured in transfection experiments; however, such markers can become unreliable due to being affected by resource loading[33,35]. To minimize the impact of resource loading on the accuracy of measurements, we thus measured reporter outputs as the median of cells gated positive for either the reporter or the TX marker (see Supplementary Note 1 for further discussion of gating strategies). To enable conversion of Gal4 levels to fluorescence values, we co-titrated a reporter (Gal4 marker) with the Gal4 TAs. Details about plasmid dosages and transfection reagents used in each experiment are provided in Source Data.

The resulting dose–response curves for activation of UAS: output$_2$ and knockdown of CMV:output$_1$ via resource loading are shown in Supplementary Fig. 2 and Fig. 1f, respectively (see also Supplementary Fig. 3 for the corresponding distributions of expression levels). At the highest dosage tested, all five Gal4 TAs knocked down CMV:output$_1$ by at least 30%, with Gal4-VPR causing ~80% knockdown (Fig. 1f). Additional qPCR and flow-cytometry measurements validated that the effect of Gal4 TAs on CMV-driven expression is caused by the ADs and occurs predominantly at the transcriptional level (Supplementary Fig. 4). Consistent with prior studies[29,31], the activation dose–response curve of some Gal4 TAs (Gal4-Rta, Gal4-p65, and Gal4-VPR) showed decreasing UAS:output$_2$ at high dosages of the TAs, presumably due to self-squelching (Supplementary Fig. 2a–c).

We developed a mathematical model of gene expression that accounts for transcriptional and translational resources shared among genes (described in detail in Supplementary Note 2 and Supplementary Fig. 5). This model recapitulates the trends of both non-target gene knockdown (Fig. 1f) and on-target

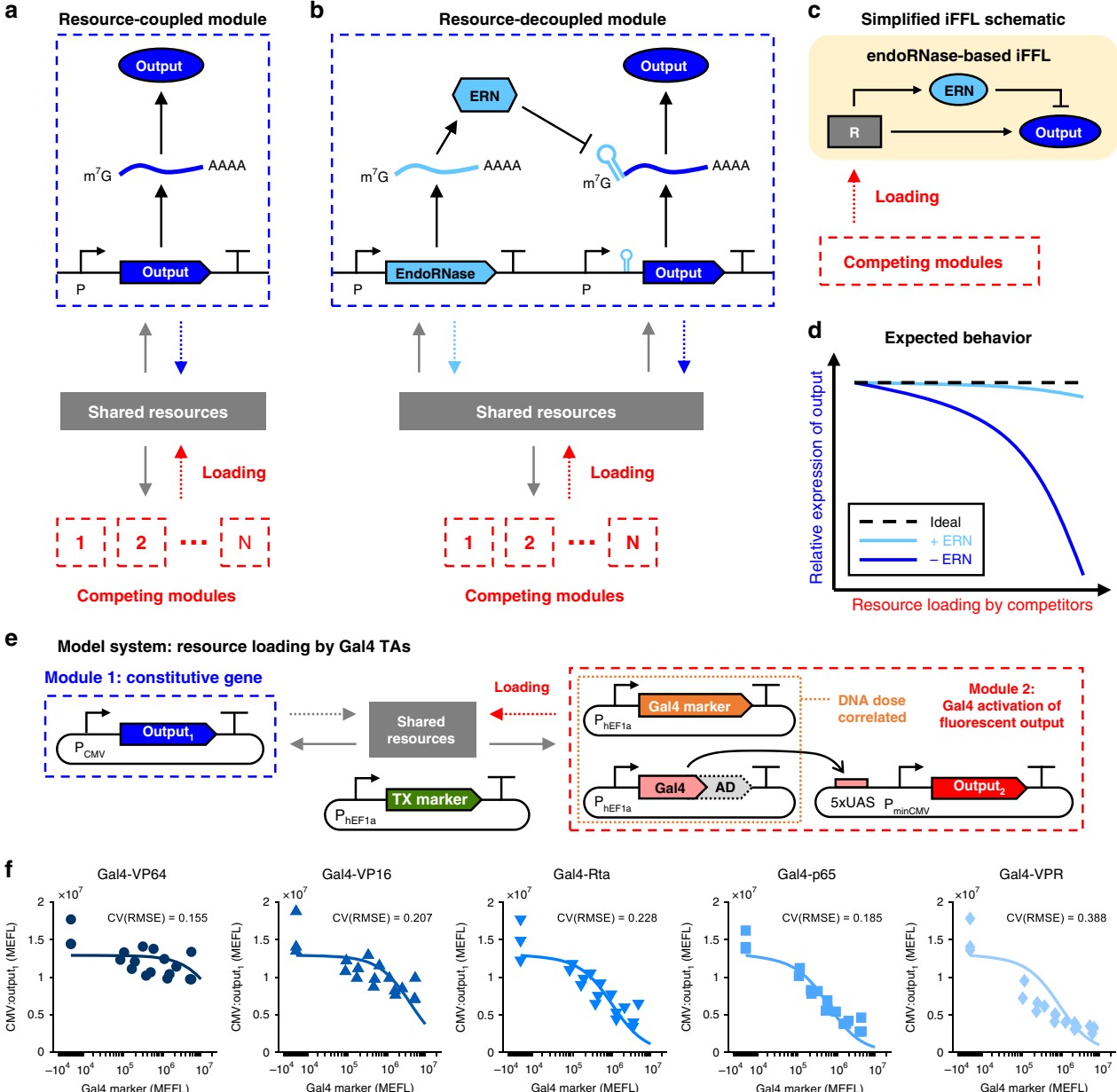

**Fig. 1 Feedforward control strategy to decouple modules that share limited resources. a** A genetic module comprising a single constitutive transcription unit. Other competing modules place a load (indicated by the dashed red arrow) on the free cellular resources, affecting the expression of the module of interest (resource-coupled module). The module of interest also applies a load to the resources (indicated by the dashed blue arrow). **b** An incoherent feedforward loop (iFFL) device within the module of interest decouples the module's output from resource variability. An endoribonuclease (endoRNase/ ERN), produced by an identical promoter as that of the output, represses the output by binding to a specific target site in its 5′ untranslated region (UTR) and cutting the mRNA. **c** A simplified schematic of the iFFL showing cellular resources (*R*) as a disturbance input to the iFFL. **d** The expected behavior of the output of the resource-coupled (−ERN) and resource-decoupled (+ERN) modules in response to resource loading by other modules. **e** Experimental model system to recapitulate resource loading. The module of interest comprises a constitutively expressed protein (output$_1$, mKate2). In a competitor module, Gal4 transcriptional activators (TAs) drive expression of another protein (output$_2$, EYFP). Different activation domains (ADs) were fused to the DNA-binding domain (DBD) of Gal4. A reporter (Gal4 marker, TagBFP) was titrated together with the Gal4 TAs to mark their delivery per cell. The TX marker is iRFP720. **f** Dose-dependent effect of Gal4 TAs on output$_1$ for different ADs (VP64, VP16, Rta, p65, VPR). The markers indicate median expression levels from three experimental repeats. The lines represent fits of our resource competition model (equation (50), see Supplementary Note 2 and Supplementary Fig. 5). Dose–response curves and model fits for output$_2$ are shown in Supplementary Fig. 2. The CV(RMSE) is the root-mean-square error between the model and data, normalized by the mean of the data. All data were measured by flow cytometry at 48 h post transfection in HEK-293FT cells. All measurements were made on cells gated positive for the transfection marker (TX marker) or output$_1$, and are shown separately for each of three experimental repeats. MEFLs are calibrated flow-cytometry units as described in "Methods". Median values and fit parameters are provided in Source Data.

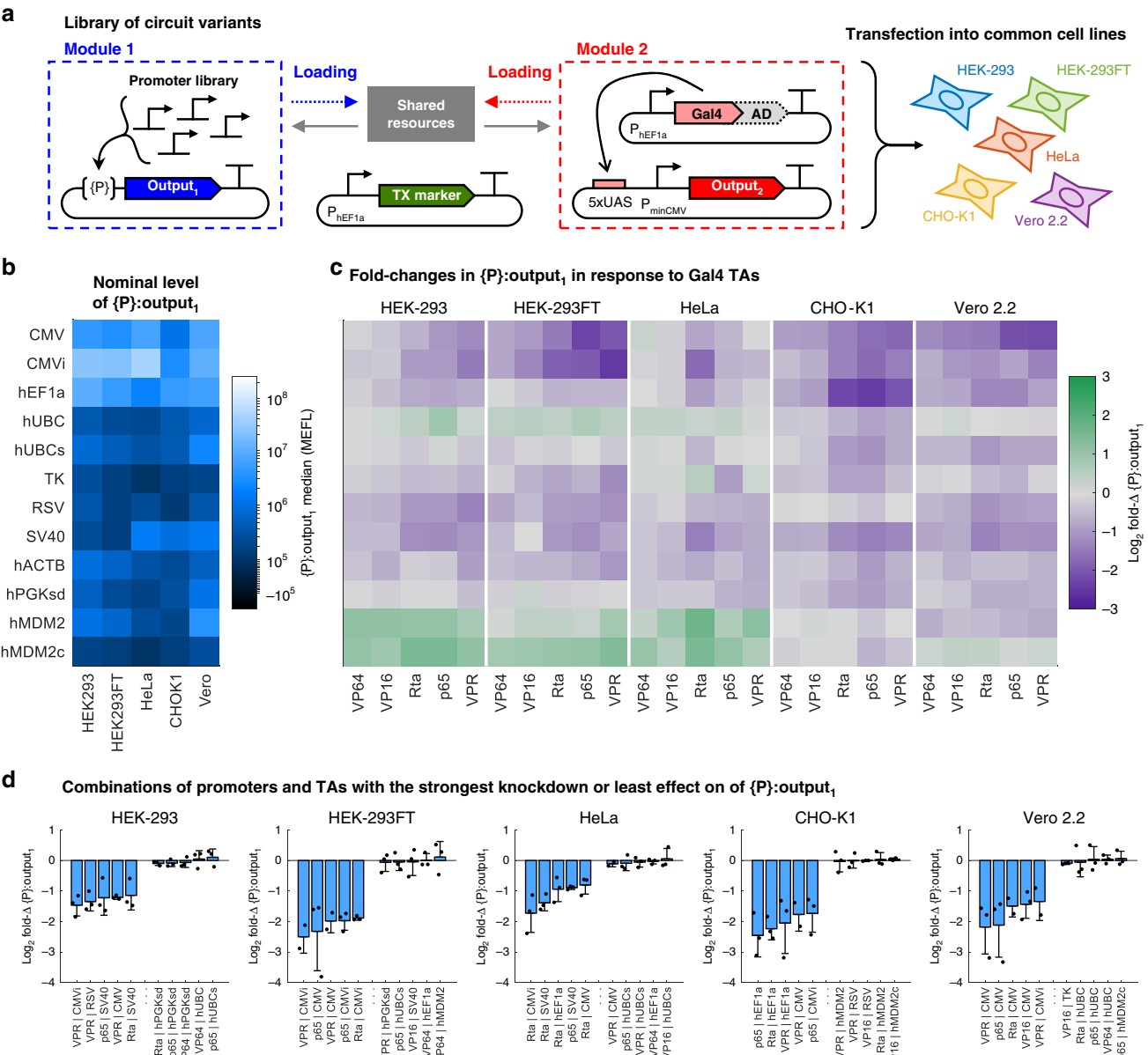

**Fig. 2 Effect of resource competition between promoters and activators across cell lines. a** Genetic model system to study competition for resources between different combinations of constitutive promoters ({P}:ouptut$_1$, mKO2) and Gal4 TAs driving a second output (UAS:output$_2$, TagBFP) in different cell lines. The specific promoters, ADs fused to Gal4, and cell lines are shown alongside the data in panels (**b**, **c**). A description of each constitutive promoter is provided in Supplementary Table 1. The TX marker is mNeonGreen. **b** Nominal outputs are the median expression levels of each promoter in Module 1 in each cell line when co-transfected with Gal4-None (i.e., the Gal4 DNA-binding domain), which does not load resources (Supplementary Figs. 1 and 4). **c** Fold changes (fold-Δs) in the level of {P}:output$_1$ in response to Gal4 TAs. The fold-Δs are computed by dividing the median level of {P}:output$_1$ for each constitutive promoter when co-transfected with different Gal4 TAs, by the nominal output of the promoter in the same cell line. **d** The five promoter–activator combinations in each cell line with the smallest effect or largest negative effect on the level of output$_1$. Error bars represent the mean ± s.d. of measurements from three experimental repeats (which are represented by the individual points). All data were measured by flow cytometry at 48 h post transfection in the cell lines indicated. Panels (**b**, **c**) show the mean of measurements from three experimental repeats. All measurements were made on cells gated positive for TX marker or {P}:output$_1$. Median values for each reporter in each sample are shown in Supplementary Fig. 11 and provided in Source Data.

self-squelching by TAs observed in the experiments (Supplementary Fig. 2b). For further discussion of model fitting and validation, see Supplementary Note 3 and Supplementary Figs. 6–8. Importantly, the qualitative trends displayed by the model were also predictive of circuit behavior in lentiviral-integrated contexts (Supplementary Figs. 9 and 10), indicating that the qualitative effect of resource loading by TAs, i.e., decrease in the expression of non-TA target genes, apply to genes located in both plasmids

and chromosomes (for further discussion, see Supplementary Note 4).

To determine how resource loading affects different constitutive promoters and whether the cellular host modulates these effects, we carried out the experiment shown in Fig. 2a and Supplementary Fig. 11. In particular, we extended our model system from Fig. 1e to test the effect of different Gal4 TAs on a library of non-target constitutive promoters in Module 1 ({P}:

output₁—see Supplementary Table 1 for more details) when transfected into various commonly-used cell lines. Figure 2b shows the nominal expression levels (i.e., the median expression level in the absence of resource loading, measured in this experiment using samples co-transfected with the Gal4 DBD (no AD)—see "Methods") of {P}:output₁ in Module 1 for each constitutive promoter in each cell line tested. We then computed fold changes in response to each Gal4 TA by normalizing the median expression level of {P}:output₁ in each combination of a constitutive promoter, TA, cell line to the nominal expression level of the same constitutive promoter in the given cell line and in the absence of the TA (Fig. 2c).

From the {P}:output₁ fold changes, we can extract patterns that help guide design choices for genetic circuits. Decreased expression of {P}:output₁ was observed in the majority of combinations, with viral promoters being generally more negatively affected by resource loading than human promoters (see also Supplementary Fig. 12). While the relative effects of Gal4 TAs on each constitutive promoter were reasonably correlated between cell lines (0.5 < r < 0.9), the exact fold changes were poorly predictable between one cell line and another (Supplementary Fig. 13). The differences among cell lines may result from the promoters utilizing distinct subsets of transcriptional resources[52–55] that are differentially loaded by each TA and differentially expressed within each cell line. We observed appreciable increases in output for three of the twelve promoters tested (hUBC, hMDM2, and hMDM2c— Fig. 2c). Several of the TAs including Gal4-Rta, -p65, and -VPR were observed to cause reductions in cell division rate as measured by Ki-67 staining (Supplementary Fig. 14). Accounting for changes in growth rates in simulations, along with analysis of cell density in the experimental data, suggest that decreases in cell division rate due to toxicity of the TAs may explain the increase in expression of hUBC and hMDM2c promoters (see Supplementary Note 2 and Supplementary Figs. 15 and 16). However, the increase in output expression for the full-length hMDM2 promoter appears to be Gal4-specific and not correlated with changes in cell density (Supplementary Figs. 16 and 17). The presence of two consensus Gal4-binding sites in the hMDM2 promoter sequence (see the sequence in Source Data) suggests that Gal4 TAs can bind and activate transcription of hMDM2 (Supplementary Note 3).

While we saw widespread reductions and in some cases increases in {P}:output₁ in response to the Gal4 TAs, there were some combinations of promoters and Gal4 TAs in each cell line that had little to no effect. The five promoter–TA combinations with either the strongest knockdown of or least effect on output₁ are reported in Fig. 2d (see Supplementary Fig. 18 for all combinations). In particular, the hUBC and hPGK promoter variants were frequently found to be unaffected by the Gal4 TAs. However, individual combinations of constitutive promoters and Gal4 TAs that are relatively uncoupled in one cell line are not generally uncoupled in different cell lines. Only three combinations that showed the least coupling in one cell line (VP64/hEF1a, VP64/hUBCs, and p65/hUBCs) were shared among at least two different cell lines. Therefore, while in individual cell lines it is possible to find combinations of genetic parts that result in reduced coupling due to resource sharing, a general method that is agnostic of the specific genetic parts used and is applicable to any given cell lines is needed to decouple gene expression from competition for shared resources.

**Design of an endoRNase-based feedforward controller**. In order to mitigate the effect of resource loading on any genetic module's output, we designed a resource-decoupled genetic module by augmenting Module 1 with a feedforward controller (Fig. 1b).

The feedforward path of the controller is obtained by expressing an endoRNase that targets the output protein's mRNA for degradation. The promoter expressing the endoRNase is identical to that expressing the output, ensuring that the expression of both genes depends on the same transcriptional and translational resources. This controller architecture leads to an incoherent feedforward loop (iFFL) motif (Fig. 1c). Qualitatively, with reference to Fig. 3a, as the availability of transcriptional or translational resources ($R$) decreases, such as due to loading by TAs, the level of the endoRNase ($x$) also decreases, de-repressing the output protein ($y$). If the system is properly designed, this action should compensate for the decrease in output production caused by a decrease in available resources, thus enabling the level of the output protein to remain unchanged for a range of perturbations in the resource amount $R$.

The extent to which the output level remains unchanged (i.e., the robustness of the iFFL design) is dependent on a number of biochemical parameters. To extract the key tunable parameters dictating the robustness of this iFFL design, we use a mathematical model based on mass-action kinetics (see "Methods" and Supplementary Note 5 for derivation). According to this model, the steady-state output protein level $y$ of the iFFL is given by:

$$y = V_y \cdot \frac{D \cdot R}{1 + D \cdot R/\epsilon}, \qquad (1)$$

where $R := R_{TX} \cdot R_{TL}$ lumps together the free concentrations of transcriptional ($R_{TX}$) and translational ($R_{TL}$) resources, and $D$ is the concentration of the DNA plasmid that encodes both the output and the endoRNase. The lumped parameters $V_y$ and $\epsilon$ are defined as:

$$V_y := \frac{\varphi_y \beta_y}{\gamma_y k \kappa_y \delta_y}, \quad \text{and} \quad \epsilon := \frac{\gamma_x k \delta_x \delta_y K_M}{\varphi_x \beta_x \theta} \cdot \kappa_x, \qquad (2)$$

where, for $i = x$ (endoRNase) or $i = y$ (output), $\varphi_i$ is the transcription initiation rate constant; $\delta_i$ is the decay rate constant of the mRNA transcript $m_i$; $\gamma_i$ is the decay rate constant of protein $i$; $\beta_i$ is the translation initiation rate constant; and $\kappa_i$ is the dissociation constant describing the binding between translational resource (i.e., ribosome) and the mRNA transcript $m_i$, and thus governs translation initiation. The parameter $\theta$ is the catalytic rate constant of the endoRNase cleaving $m_y$, $K_M$ is the Michaelis–Menten constant describing the binding of the endoRNase with $m_y$, and $k$ is the dissociation constant describing binding of transcriptional resource with the two identical promoters driving the expression of both endoRNase $x$ and output $y$. Overall, changing the biochemical parameters governing the production, decay, and enzymatic reactions of the endoRNase only changes the lumped parameter $\epsilon$, while the lumped parameter $V_y$ is entirely determined by biochemical parameters of the output gene. The derivation of Eq. (1) is independent of the resource sharing model developed in Supplementary Note 2 and is only based on the assumption that the endoRNase ($x$) and the output protein ($y$) are using the same pool of resources for transcription and translation (Supplementary Note 5). According to Eq. (1), if $D \cdot R \gg \epsilon$, then $y \approx Y_{max} := V_y \cdot \epsilon$, which is independent of $R$ and therefore independent of the free concentrations of both transcriptional and translational resources. We call the lumped parameter $\epsilon$ *feedforward impedance* because as $\epsilon \to 0$, the condition $D \cdot R \gg \epsilon$ can be more easily satisfied (i.e., it is satisfied for a wider range of $D \cdot R$). As a consequence, the feedforward control action exactly cancels out the effect of a change in resource availability ($R$) on the output, although with the trade-off of a lower output level $Y_{max}$ (Fig. 3b). The experimentally quantifiable value $Z_{50} = V_z \cdot \epsilon$

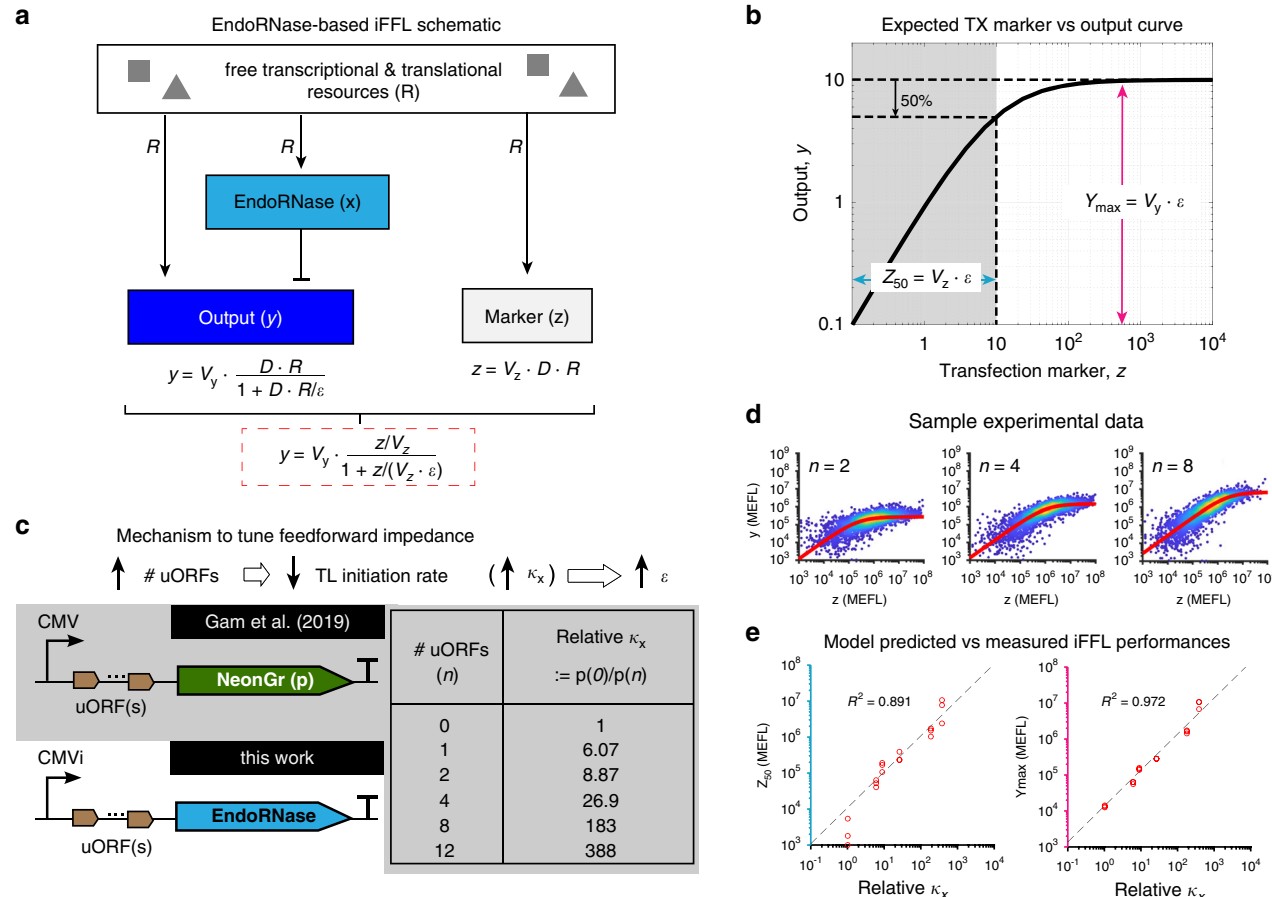

**Fig. 3 Model-guided design and tuning of the endoRNase-based feedforward controller. a** A schematic of the endoRNase-based feedforward controller. The expression of endoRNase ($x$), output ($y$), and TX marker ($z$) all use the same pool of transcriptional and translational resources ($R$). Parameters $V_y$ and $\epsilon$ are defined in Eq. (2). Parameter $V_z$ is defined in Eq. (85) in Supplementary Note 5. **b** The TX marker ($z$) vs output ($y$) dose–response curve computed from the model in Eq. (1). See "Methods" for derivation. The shape of this curve can be characterized by $Y_{max}$ and $Z_{50}$, both of which are proportional to the feedforward impedance $\epsilon$. **c** An increase in the number of upstream open-reading frames (uORFs) in the 5′ UTR of the endoRNase transcript leads to a decrease in its translation initiation rate. We model it as an increase in the dissociation constant between the ribosome and the endoRNase's mRNA transcript ($\kappa_x$), which increases $\epsilon$. The relationship between the number of uORFs and the fold decrease in translation initiation (i.e., parameter $\kappa_x$ in the model) is summarized in the table using previously-published experimental data by Gam et al.[44]. **d** Sample experimental data (scatterplot) corresponding to $n = 2, 4, 8$ overlaid with TX marker vs output model fitting (red solid line). $n$ is the number of uORFs in the 5′ UTR of the Cas6-family endoRNase CasE (EcoCas6e). Experimental data are excerpted from Fig. 6b. **e** Comparison between experimentally measured $Z_{50}$ and $Y_{max}$ and the relative difference in ribosome–mRNA dissociation constant ($\kappa_x$) for different numbers of uORFs in the 5′ UTR of the endoRNase. The experimental data are fit to Eq. (3) to extract $Z_{50}$ and $Y_{max}$ (see "Methods"). Fit parameters are provided in Source Data.

is the TX marker ($z$) level above which the output is at least 50% of $Y_{max}$. $Z_{50}$ can thus be regarded as an inverse measure of the iFFL's robustness to changes in resource availability (Fig. 3a, b).

To achieve a system with low feedforward impedance, we implemented the controller with Cas6-family CRISPR endoR-Nases[46]. These endoRNases bind to and cleave specific ~20–30 bp-long hairpins in RNA sequences (independent of guide RNAs), yielding between ~50-fold and 250-fold knockdown of target proteins[46], indicating a high catalytic rate $\theta$ to reduce $\epsilon$ according to Eq. (2). Of these, we chose CasE[45], one of the endoRNases with the highest gene knockdowns that we have evaluated[46]. We placed the target site for CasE in the 5′ UTR of the output gene's transcript because Cas6-family endoRNases more strongly knock down gene expression when targeting the 5′ UTR than when targeting the 3′ UTR[46,56]. To construct a library of CasE iFFLs with different feedforward impedance ($\epsilon$), we placed variable numbers of upstream open-reading frames (uORFs)[57] in the 5′ UTR of the CasE transcription unit, thereby

varying the translation rate of CasE (Fig. 3c). This method of tuning the feedforward impedance, as opposed to using different-strength promoters, allows the promoters driving CasE and the output to be identical, ensuring that both genes use the same pool of transcriptional resources. In this scheme, increasing the number of uORFs ($n$) effectively increases the dissociation constant $\kappa_x$ between the ribosome and $m_x$[57] to decrease $\epsilon$ (Fig. 3c).

We experimentally verified this model prediction for $n = 0, 1, 2, 4, 8$, and 12. Regardless of the number of uORFs, the shapes of the experimentally measured TX marker vs output dose–response curves match our model well (see Fig. 3d for select samples). Variants of the iFFL with fewer uORFs yield a smaller fit value of $Z_{50}$, suggesting that they will be more robust to changes in $R$ (Fig. 3d, e). Furthermore, our model predicts that $Z_{50}$ and $Y_{max}$ are both proportional to $\epsilon$ and, hence, $\kappa_x$ (Fig. 3b). Indeed, the fit values of both $Z_{50}$ and $Y_{max}$ are linear to the expected changes in $\kappa_x$ based on values from Gam et al.[44] (Fig. 3e). This implies that the number of uORFs placed on 5′

UTR of the CasE transcript can quantitatively shape the input/output response of the iFFL.

**The iFFL output adapts to resource loading**. In our genetic implementation of the iFFL, we used the CMVi promoter to drive the expression of both CasE and the output (Fig. 4a). We chose the CMVi promoter because it is strongly knocked down by Gal4 TAs across cell lines (Fig. 2), thus providing an ideal test bed for assessing the controller performance. To evaluate the benefit of the iFFL design, we made an unregulated (UR) variant of Module 1 (Fig. 4a) that replaces CasE with the luminescent protein Fluc2, thus breaking the feedforward path. To measure the response of the iFFL and UR modules to resource loading, we co-transfected plasmids encoding variants of them along with plasmids expressing a TX marker and hEF1a:Gal4-VPR into HEK-293FT cells. To account for differences in protein expression levels between the UR and iFFL modules, we transfected cells with equimolar, 1:4, 1:16, or 1:64 dilutions of the UR plasmid relative to the amount of iFFL plasmid used for iFFL variants.

As predicted by the model, our experimental results show that variants of the iFFL with fewer uORFs are more robust to changes in resource availability (Fig. 4b, c and Supplementary Fig. 19). Fold changes and robustness scores were computed relative to the samples without Gal4-VPR for each UR and iFFL device independently (see "Methods"); the maximum robustness score is 100%. At the highest dosage of Gal4-VPR tested (30 ng), the output of the UR samples decreased between twofold and threefold, whereas the iFFL variants with 4× or 2× uORFs were nearly unaffected (Fig. 4b). In terms of robustness scores, most UR samples had a score of ~30–50% regardless of the nominal output level, whereas iFFL variants with 4× or 2× uORFs had a score of ~70–90% (Fig. 4c). iFFL variants with increasing numbers of uORFs (up to 12×) have robustness scores that approach those of the UR samples. To ensure that the superior performance of iFFL variants with fewer uORFs did not result from reduced sensitivity to measuring lower output levels, we directly compared UR and iFFL variants with similar nominal output levels (1:64 diluted and 4× uORFs, respectively). Whereas the UR/64 output decreases by ~60% and its distribution clearly shifts down in response to resource loading by Gal4-VPR, the iFFL output is nearly unchanged and its distribution retains approximately the same median with comparable variance (Fig. 4d, e). Overall, these data validate the model prediction that decreasing $\epsilon$ increases robustness to resource loading, but has a trade-off in reducing the output level.

According to the model of our iFFL, $Y_{max} = V_y\epsilon$; thus, in order to increase the iFFL output level without changing robustness ($\epsilon$), we can increase $V_y$. This can be achieved by increasing the transcription or translation rates or by decreasing the decay rate of the output protein. To validate this prediction, we measured the iFFL output at various ratios of CasE and output plasmids using poly-transfection[44]. Indeed, increasing the output DNA dosage (and thus transcription rate) relative to that of CasE increases the fit value of $Y_{max}$ while maintaining an approximately constant fit value of $Z_{50}$ (Supplementary Fig. 20).

We next tested whether the iFFL module functions in other cell lines and whether its output expression is robust to resource loading by different Gal4 TAs (Fig. 5a and Supplementary Fig. 21). In these experiments, we added higher dosages of Gal4 TAs than in Fig. 4 to challenge the iFFL with high loading conditions (see Source Data for transfection tables). Overall, we found that fold changes in output of the iFFLs in response to resource loading are much lower than those in comparable UR systems for nearly all combinations of Gal4 TAs and cell lines tested (Fig. 5b, c and Supplementary Figs. 22–24). Directly comparing UR and iFFL variants with similar nominal output levels (UR/10 vs 8xU-CasE and UR/100 vs 4xU-CasE) in each cell line, we observed that the iFFL is able to maintain the desired output level even when the UR output is strongly reduced (Fig. 5d). Specifically, in situations where the UR device's output was affected by more than 30% ($-0.5\log_2$ fold-Δ output₁), the iFFL device's output was typically unaffected and rarely affected to the same degree. Moreover, in combinations where the output of the UR device was highly affected (up to 70% in HeLa and U2OS), that of the iFFL was only slightly affected (unappreciable change in the 4xU variant and less than 30% in the 8xU variant with larger $\epsilon$ and higher output). Across cell lines, the robustness scores of the iFFL variants were nearly always higher than those of the UR variants (Supplementary Fig. 22a–d). Most strikingly, the percent of samples with robustness scores over 80% in HeLa, CHO-K1, and U2OS cells increased from 31%, 8.9%, and 20% for UR variants to 100%, 84%, and 93% for iFFL variants, respectively (Supplementary Fig. 22e). Thus, even in cell lines in which unregulated genetic devices exhibit high sensitivity to resource loading (Fig. 2), our iFFL design can substantially reduce the effects of resource loading on gene expression.

To ensure that our results were not specific to the CMVi promoter, we repeated the experiments above using a version of the iFFL that replaces the CMVi promoters with the hEF1a promoter (Supplementary Figs. 25–29). As in the CMVi iFFL, variants of the hEF1a iFFL with fewer uORFs/lower output generally showed reduced fold changes and higher robustness scores in response to Gal4 TAs than UR variants with comparable nominal outputs (Supplementary Figs. 25 and 27). Compared to the CMVi iFFL, the hEF1a iFFL generally showed higher fold changes and lower robustness scores, especially in U2OS and HeLa cells co-transfected with Gal4-Rta (Supplementary Figs. 27–29). For hEF1a iFFL variants with 4x or fewer uORFs, the output level was increased by the Gal4 TAs in HEK-293 and HEK-293FT cells. This increase can be attributed to the toxicity of the Gal4 TAs that can be lessened by using a less toxic transfection reagent (see Supplementary Note 5 and Supplementary Figs. 5, 30–33). Notably, for both the CMVi and hEF1a iFFLs, the nominal output levels for variants with different numbers of uORFs were highly correlated across cell lines (Supplementary Figs. 34 and 35), suggesting that the iFFL also generally mitigates the effects of contextual differences between cell lines, such as the overall abundance of gene expression resources.

**The iFFL output adapts to plasmid DNA copy number variation**. Following from previous work with miRNA- and transcriptional repressor-based iFFLs[58–60] and from the model of our endoRNase-based iFFL design (Fig. 3), we predicted that the output level of our iFFL module would also be robust to variation in its DNA copy number (see "Methods"). We thus tested whether, absent resource loading, output expression of the hEF1a iFFL could adapt to the multiple log decades of variation in plasmid uptake between individual cells seen in transient transfections (Fig. 6a). As the level of a TX marker is proportional to DNA copy number[58], we were able to use the TX marker vs iFFL output curves to fit Eq. (3) (see "Methods"), and found good agreement between the data and model (Fig. 6b). Binning of cells at different TX marker levels (and thus DNA dosages) shows that the level of iFFL output indeed becomes insensitive to the plasmid copy number of the iFFL above a minimal amount of DNA input (Fig. 6b). Similar binning analysis for UR variants indicates that simply decreasing output expression does not cause adaptation to DNA copy number (Supplementary Fig. 36a). To quantify the extent of iFFL output adaptation to DNA copy number, we compared the median expression of cells in TX marker-delineated

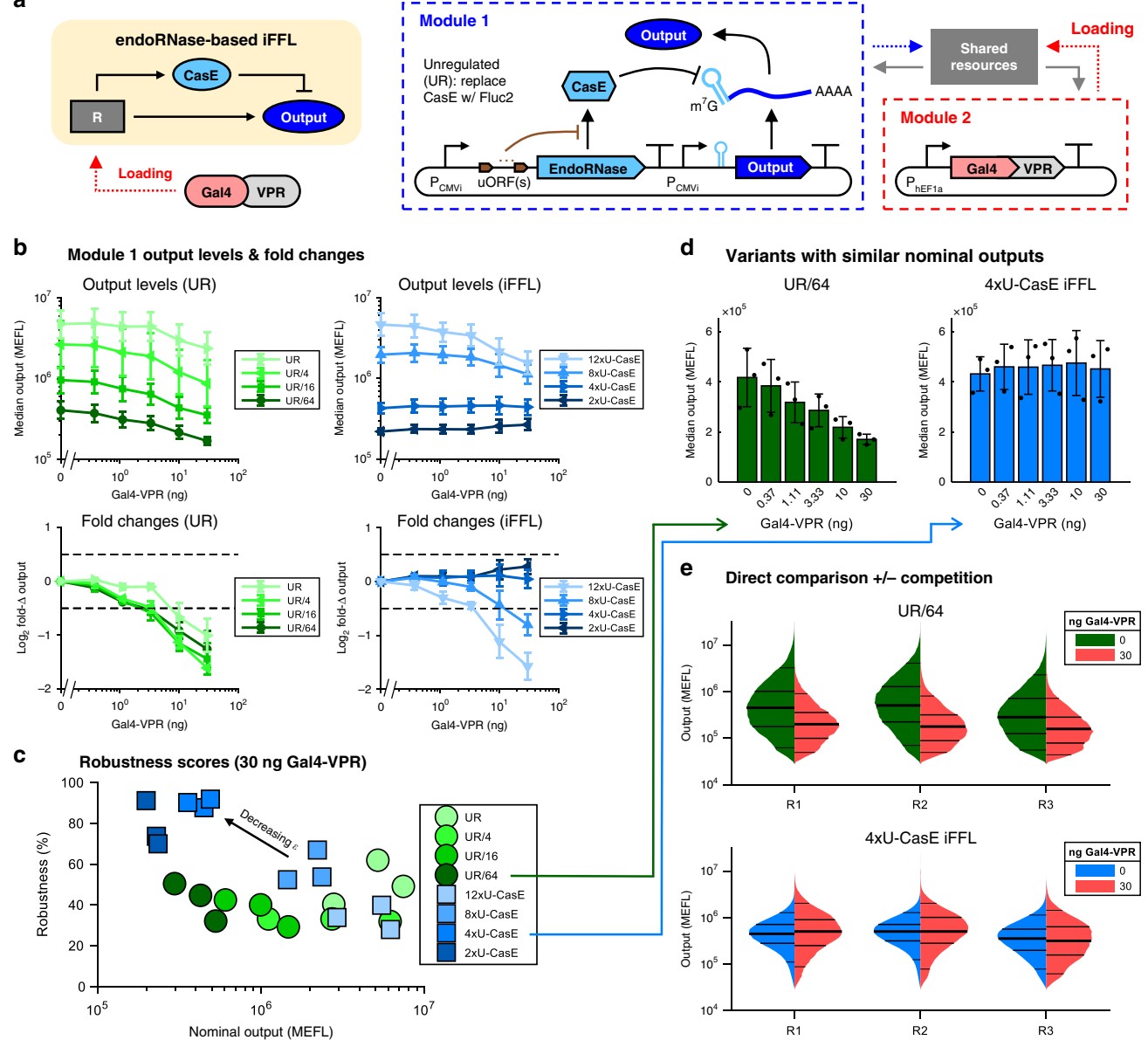

**Fig. 4 Robustness of the iFFL output level to resource loading by Gal4-VPR. a** Schematic and genetic diagram of the endoRNase-based iFFL module. 0-12 uORFs are placed in the 5′ UTR of CasE mRNA to reduce its translation rate ('#xU-CasE': # ≡ number of uORFs). CasE binds and cleaves a specific target site in the 5′ UTR of the output mRNA. The unregulated (UR) version of Module 1 replaces CasE with Fluc2, has no uORFs, and retains the CasE target site in the output mRNA. Samples with reduced UR plasmid copy numbers ('UR/#': # ≡ reduction factor) are provided to compare UR and iFFL variants with similar output levels. The output reporter is EYFP; not shown are two constitutive reporters (CMVi:TagBFP, hEF1a:mKO2). **b** Response of iFFL module to resource loading by Gal4-VPR. The top plots show the mean ($\mu$) ± relative error ($\frac{1}{\ln(10)} \cdot \frac{\sigma}{\mu}$) of median output levels. The relative error is used to more accurately represent error on the log-scale[76]. The bottom plots show the mean ± s.d. of fold changes (fold-Δs). Fold-Δs are computed by dividing the median level of output at a given dosage of Gal4-VPR by that at 0 ng Gal4-VPR (i.e. the nominal output level—see Eq. (6) in "Methods"). Data are from three experimental repeats. **c** Robustness scores (100% minus % deviation from the nominal level—see Eq. (8) in "Methods") computed for each UR and iFFL variant at 30 ng Gal4-VPR. Symbols represent measurements from each experimental repeat. **d** Direct comparison of UR and iFFL variants with similar nominal outputs. Error bars represent the mean ± s.d. of measurements from three experimental repeats. **e** Distributions of output levels per cell at 0 and 30 ng Gal4-VPR for the UR and iFFL variants shown in panel (**d**), for all three experimental repeats (R#). The lines on the histograms denote the 5th, 25th, 50th, 75th, and 95th percentiles. All data were measured by flow cytometry at 72 h post transfection in HEK-293FT cells. All measurements were made on cells gated positive for output. Measurements on cells gated positive for either output or TX marker are shown in Supplementary Fig. 19. iFFL samples with 0 or 1 uORFs are not shown because most or all of the cells in those samples did not express output above the autofluorescence background and their median expression levels were much lower than that of any UR variants. Median values for each sample and the number of cells plotted per histogram are provided in Source Data.

bins to the fit value of the iFFL model parameter $Y_{max}$ (Supplementary Fig. 36b). We considered a bin to be adapted to DNA copy number variation if $\log_{10}$(output) was within 5% of $\log_{10}(Y_{max})$ (i.e., the log-scale robustness score was above 95%). As predicted from the model, increasing the number of uORFs

(and thus increasing the output level) decreases the range of DNA copy numbers over which the iFFL output adapts to DNA copy number variation (Fig. 6c). We repeated these experiments and analyses with the CMVi-driven CasE iFFL and found similar results (Supplementary Fig. 37).

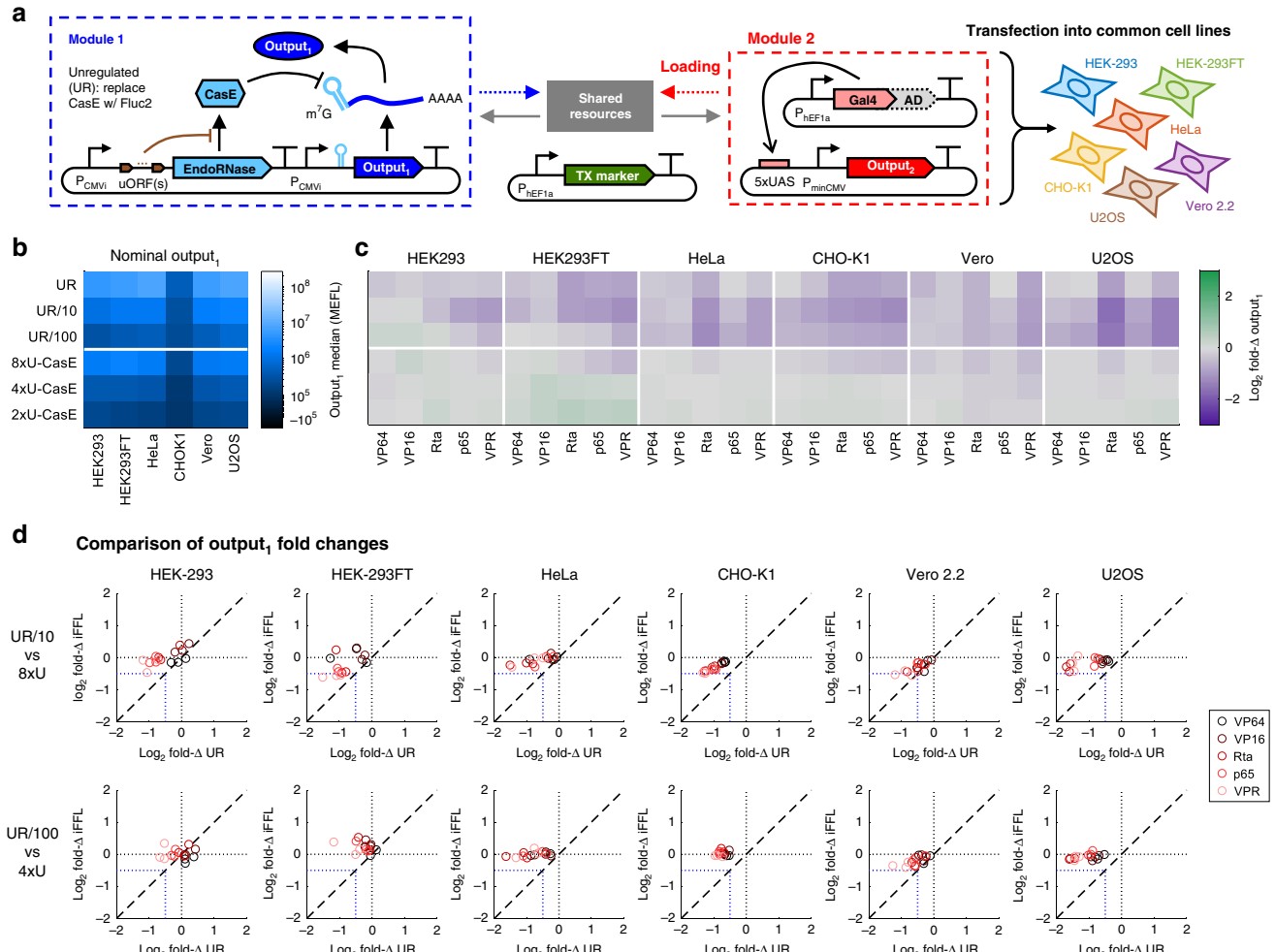

**Fig. 5 Robustness of the iFFL output level to resource loading across cell lines. a** Schematic of the experiment to test the performance of the iFFL to robustly control the level of output (output₁, EYFP) in different cell lines with different Gal4 TAs loading resources and driving expression of output₂ (TagBFP). The TX marker is mKO2. **b** Nominal outputs are the median expression levels of each UR or iFFL variant in each cell line when co-transfected with Gal4-None (i.e., the Gal4 DNA-binding domain), which does not load resources (Supplementary Fig. 4). **c** Fold changes (fold-Δs) in the level of output₁ in response to Gal4 TAs. The fold-Δs are computed independently for each UR and iFFL variant and cell line by dividing the median level of output₁ for each sample co-transfected with different Gal4 TAs by the nominal output. **d** Comparison of fold-Δs in output₁ expression in response to different Gal4 TAs in each cell line, between UR and iFFL devices with similar nominal output levels. Measurements from each experimental repeat are shown separately. The blue dotted lines indicate −0.5 log₂ fold-Δ ( ~30% decrease). All data were measured by flow cytometry at 72 h post transfection in the cell lines indicated. Panels (**b**, **c**) show the mean of measurements and from three experimental repeats. All measurements were made on cells gated positive for output₁ only. Measurements on cells gated positive for either output₁ or TX marker are shown in Supplementary Figs. 21 and 23. See Supplementary Fig. 22 for additional analysis. Median values for each reporter in each sample are shown in Supplementary Fig. 21 and provided in Source Data.

Previously, miRNA-based iFFLs[58,59,61] placed the miRNA target sites in the 3′ UTR, whereas we placed the CasE target site in the 5′ UTR. To test whether the choice of target site placement affects iFFL performance, we compared variants of our CasE iFFL, and a miR-FF4 iFFL based on the design by Bleris et al.[58], with either 5′ or 3′ target sites (Supplementary Fig. 38a). We found that for both the miRNA- and endoRNase-based iFFLs, variants with 5′ target sites show substantially improved robustness to DNA copy number variability compared to variants with 3′ target sites (Supplementary Fig. 38b, c). Thus, the location of the target site an important design choice for iFFLs with mRNA-targeting regulators.

We further investigated whether the iFFL could also adapt to temporal variation in DNA copy number. This problem occurs during transient transfections because plasmids are diluted out with cell division, causing output expression to decrease with time and complicating measurements. Our model suggested that the

iFFL module could maintain the output expression level for a longer period of time compared to UR samples (see "Methods" and Supplementary Note 5). Indeed, variants of the iFFL with fewer uORFs (and thus smaller $\epsilon$) exhibited decreasing changes in median expression over the time period of 120 h post transfection (Fig. 6d, see Supplementary Fig. 39 for full distributions at each time point). To provide a reference for our iFFL's dynamics, we compared it to the miR-FF4 iFFL with 5′ target sites. Even though the maximum output level of the miR-FF4 iFFL was similar to that of the 4x-uORF CasE iFFL, the output level of the former varied substantially more over time (Fig. 6d). Specifically, the output level of the miRNA-based iFFL initially increases by ~50% from 12 to 24 h and then decreases by ~85% from 24 to 120 h, whereas that of the best performing endoRNase-based iFFL (1xU-CasE) does not change from 12 to 24 h and decreases by only ~50% from 24 to 120 h. Simulations of the iFFL during transient transfection indicate that the ability of the iFFL to adapt

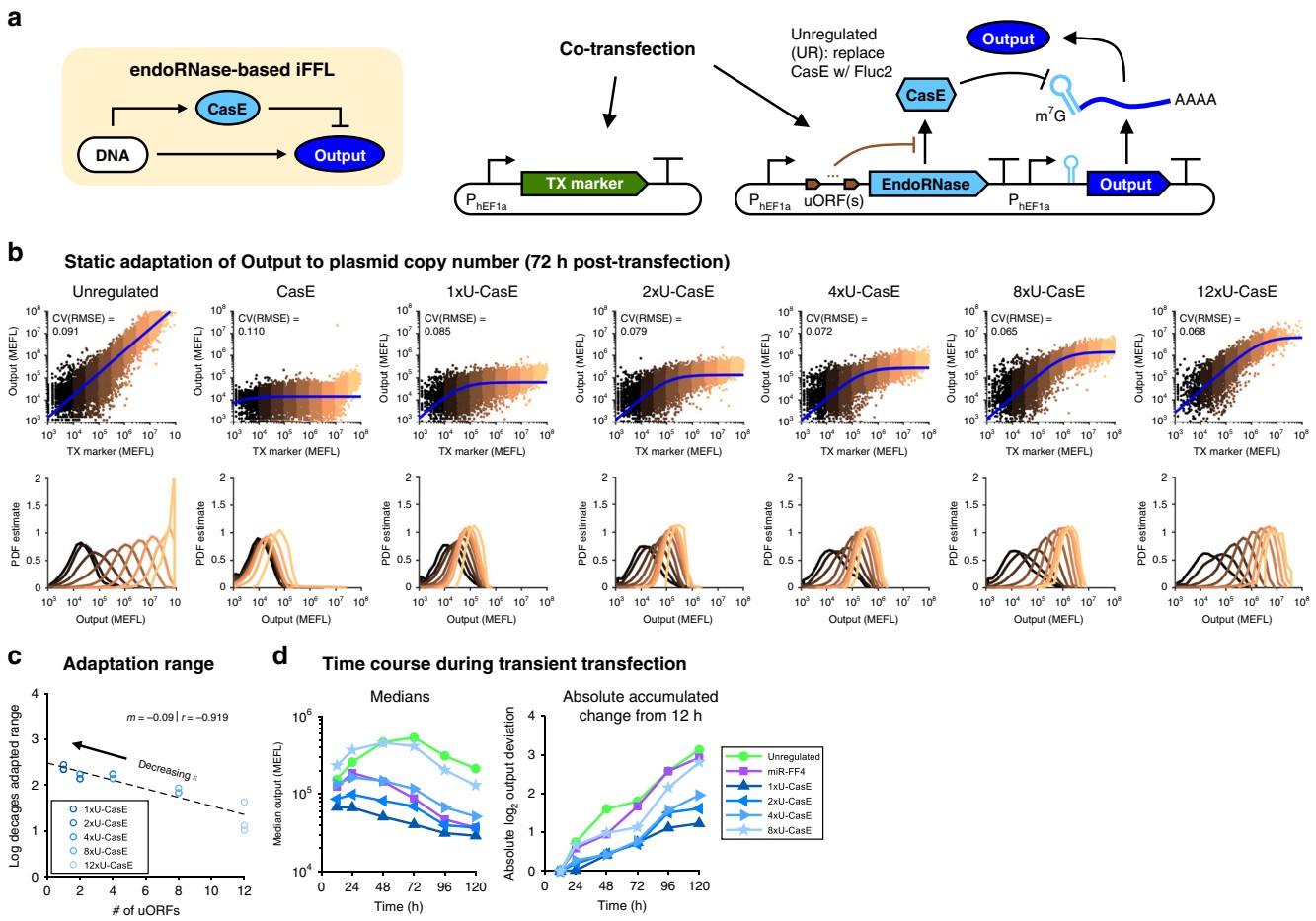

**Fig. 6 Adaptation of the iFFL output level to DNA copy number variation. a** Genetic diagram of the iFFL with hEF1a promoters, showing DNA as the input. No resource competitor was added in this experiment. The output is mNeonGreen. The constitutive TX marker (TagBFP) reports the plasmid dosage delivered to each cell. **b** Top row: TX marker vs output levels for each sample, overlaid with fits of the iFFL model. For the UR samples, the output is proportional to the TX marker, so we fit with a simple linear formula: output $= m \cdot$ TX marker. The CV(RMSE) is the root-mean-square error between the model and non-binned data, normalized by the mean of the data ($\log_{10}$-transformed first since the cell-to-cell variance is approximately log-normally distributed). To facilitate better comparability among plots, each bin was sub-sampled with the same number of cells ($n = 3000$). Bottom row: histograms of the output levels for cells within each color-coded bin (as indicated in the scatters). Data are representative and taken from the first of three experimental repeats. **c** Correlation between the range of DNA copy numbers over which the output of an iFFL variant is adapted and the number of uORFs in the 5′ UTR of CasE's transcription unit. The adaptation range is defined as the largest sum of the log widths of contiguous adapted bins in a sample (individual bins shown in Supplementary Fig. 36). Individual experimental repeats are shown separately. **d** Median expression over time for UR and iFFL variants (including a 5′ UTR-targeted miRNA-based iFFL for comparison— see Supplementary Fig. 38). The absolute accumulated change is the sum of the absolute values of the $\log_2$ changes in median expression between time points, summed from 12 to 120 h. The results are from populations of cells gated positive for either output or TX marker. The iFFL variant with 0x-uORFs is omitted from panels (**c**, **d**) because its output level is nearly undetectable. Median values and fit parameters are provided in Source Data.

to plasmid dilution depends on fast production and decay rates of the endoRNase (Supplementary Fig. 40). Overall, these data demonstrate that the CasE iFFL can also accurately set gene expression levels regardless of DNA dosage to cells and in the face of dynamic transcriptional disturbances such as plasmid dilution.

## Discussion

Context dependence of genetic circuits due to factors such as resource loading is a pervasive problem that hampers our ability to engineer systems that behave as intended[5]. Therefore, approaches that aid robust, predictable, and reliable engineering of genetic circuits across various contexts are needed[2,4]. In this paper, we have demonstrated that resource loading affects many common cell lines used in mammalian synthetic biology across nearly all combinations of routinely used promoters and TAs tested (Figs. 1 and 2), pinpointing resource variability as a culprit

of circuit malfunction in mammalian cells. We designed a feedforward controller that can make a GOI's expression level robust to resource variability. Specifically, in situations where resource loading by TAs knocked down the expression level of an unregulated GOI (UR module) by up to 70%, the expression level of the feedforward-controlled GOI (iFFL module) did not show the appreciable change (Fig. 4). This indicates that our iFFL design can achieve near-perfect adaptation of ectopic gene expression in mammalian cells to changes in the intracellular context. Across combinations of six cell lines and five TAs that we tested, the output of the iFFL was consistently less affected by the TAs than that of the UR system (Fig. 5). This demonstrates that the controller is portable across cell lines and provides robustness to various resource competitors.

Near-perfect adaptation of our iFFL output to resource loading relies on decreasing the feedforward impedance $\epsilon$ (Fig. 3). In turn, reducing $\epsilon$ causes a reduction of the output level, highlighting a

trade-off between robustness and output expression (Fig. 4c). It is possible to increase the level of the iFFL output without compromising robustness by increasing the production rate of the output protein (Supplementary Fig. 20). In future designs, this may be accomplished by using expression-boosting sequences like WPRE[62,63] for the output gene. In this work, we tuned $\epsilon$ by tuning the production rate of CasE via uORFs, which reduce translation initiation[57]. According to the model, it is also possible to tune $\epsilon$ by tuning the transcription rate, catalytic efficiency ($\theta/K_M$), or degradation rate of CasE. However, tuning $\epsilon$ with uORFs is preferable to these options. Changing the promoter of the endoRNase may decouple the resource pool used by the endoRNase gene from that used by the output gene, thereby potentially reducing the ability to offset changes in resource availability. Mutating either the target site or endoRNase to reduce their binding affinity or the catalytic rate of the endoRNase does not yield easily-predictable outcomes. In addition, mutations of the target site may alter the spontaneous degradation rate of the output mRNA, thus affecting system performance. Finally, tuning the degradation rate of the endoRNase can also affect the dynamics of its expression and thus the dynamics of the iFFL output (Supplementary Fig. 40). By contrast, the use of uORFs retains resource coupling between the endoRNase and output, enables predictable tuning of the model parameters (Fig. 3), and does not directly affect the output mRNA or endoRNase dynamics.

In addition to resource loading, our endoRNase-based iFFL design enables robustness of gene expression with respect to both DNA dosage and dilution of plasmid DNA during transient transfection (Fig. 6). The number of actively-transcribed plasmids per cell delivered by transfection has been estimated to range between 1 and 100 by Bleris et al.[58]. However, the three orders of magnitude of fluorescence variation of the TX marker in most of our experiments suggest a potentially larger range of copy numbers in our systems. In the face of this variability, our iFFL output can adapt to variation in DNA dosage over ~1–2 log decades, depending on the number of uORFs in the 5′ UTR of CasE. This range of adaptation is comparable to the TALER-based iFFL implemented by Segall-Shapiro et al. in bacteria[60] and is a substantial improvement compared to the current standard of miRNA-based iFFLs in mammalian cells[58]. Previous miRNA-based iFFL designs placed the miRNA target site(s) in the 3′ UTR of the output gene, rather than the 5′ UTR as we did with CasE. Our experiments show that the position of the target site is critical for both endoRNase- and miRNA-based iFFLs, with 5′ target sites yielding markedly improved adaptation to changes in DNA copy number (Supplementary Fig. 38). Our iFFL models assume that the output mRNA species is completely destroyed when cleaved by an endoRNase/miRNA. However, whereas 5′ cleavage removes the 5′ cap, which is detrimental to translation initiation[64], 3′ cleavage may leave the transcript competent for continued translation. In addition, Cas6-family endoRNases like CasE can remain tightly bound to the sequence of RNA to the 5′ side of their cleavage site and protect the bound strand from 3′ exonucleases[56]. However, this protective mechanism is not likely to be the sole cause for the observed differences, as for miRNAs, the RISC complex instead retains a moderate affinity for the sequence to the 3′ side of the cleavage site[65]. Although we did not perform a systematic experimental investigation, our mathematical model indicates that resource loading reduces the robustness of the iFFL to variability in DNA copy number. This is because loading effectively decreases $z$ for a given DNA copy number $D$ ($z = V_z \cdot D \cdot R$, see Fig. 3), thereby moving the iFFL operation towards the regime where the output is more sensitive to changes in $D$. Consistent with this model analysis, in Supplementary Fig. 19e, f, we observe a shift of points on the CMVi TX marker ($z$) vs

iFFL output curve towards the origin in response to resource loading.

Comparing an optimized miR-FF4 iFFL with 5′ target sites to our CasE iFFL variants, we found that the output level of the CasE iFFL variants was more resistant to changes in plasmid copy number over time during transient transfection (Fig. 6d and Supplementary Fig. 39). Simulations with an ordinary differential equation model of the endoRNase-based iFFL indicate that robustness to DNA dilution during transient transfection can be achieved with high production and decay rates of the endoRNase (see Supplementary Note 5 and Supplementary Fig. 40a, b), consistent with previous theoretical studies of iFFL dynamics in other contexts[66–68]. For our endoRNase-based iFFL, we observed near-perfect adaptation of output levels to resource loading for samples measured at 72 h (Figs. 4 and 5), indicating that 72 h is a conservative upper bound for the adaptation time of the circuit to perturbations. Moreover, the distribution of output levels for the endoRNase-based iFFL variants with 0–2× uORFs are consistent between 24 and 48 h post transfection (Supplementary Fig. 39), suggesting adaptation takes much less than 72 h. More detailed temporal studies will be required to accurately assess the adaptation time of our endoRNase-based feedforward controller.

Based on our results in both plasmid (Figs. 1 and 4) and lentiviral (Supplementary Fig. 9) contexts, we estimate that a gene dosage of ~3–10 DNA copies with a strong promoter will produce sufficient TA protein (depending on its AD) to cause appreciable knockdown of non-target genes. Consequently, a feedforward controller of gene expression may be required in such contexts. Because of the observed resource loading effects on lentiviral-integrated genes (Supplementary Note 4), future work will investigate the use of our iFFL in genomically-integrated contexts such as lentiviruses and landing pads[69,70].

In our experiments, we found that transcriptional and not translational resources were significant contributors to the observed loading effects (Supplementary Fig. 4). Among transcriptional resources, it was previously shown that only addition of extra mediator and not RNA polymerase or GTFs was able to rescue the effects of squelching in in vitro transcription reactions[29], indicating that CoAs, such as the mediator, are the major limiting resource for TA-driven gene expression. Our mathematical model of resource loading takes this into account (Supplementary Note 2) and reproduces the trends observed in experimental data in most combinations of cell lines and Gal4 TAs (Supplementary Note 3). Nevertheless, the resource loading model can be improved in several directions. First, our mechanistic model does not account for changes in cell growth rates caused by TAs (Supplementary Fig. 13). This is important because we observed that several TAs (especially Gal4-Rta, -p65, and -VPR) caused measurable reductions in cell density, in part due to their effects on cell growth (Supplementary Figs. 15, 16, and 30). Reduction of cell growth decreases the dilution rate of the output protein, leading to an increase in output that can potentially offset the decreased protein production rate caused by resource loading (Supplementary Note 2 and Supplementary Figs. 14 and 15). These effects should be considered in future models of resource loading along with accurate measurements of cell growth rates. Second, our resource loading model assumes that the same resource limits expression of all genes. In reality, there are hundreds of transcriptional cofactors (including CoAs and subunits of the mediator complex) that interact with native and synthetic TFs[52,53], which could be limiting for different genes. Future work may identify the transcriptional and translational resources used by specific genetic devices and the differential availability of these resources in distinct cell lines. Finally, several samples associated with three out of twelve promoters tested (hUBC, hMDM2, and hMDM2c) showed relatively

consistent increases in output levels in response to resource loading by Gal4 TAs (Fig. 2). While for the hUBC and hMDM2c promoters we could observe some correlation between the increase in output level and reduction in cell density by the TAs, we could not observe definite correlation in the case of hMDM2 (Supplementary Fig. 15). Instead, the increase caused by the hMDM2 promoter may result from direct binding of Gal4 to consensus UAS sequences in the hMDM2 promoter and activation of the normally p53-activated minimal promoter. Thus, while the observed increases in output for these promoters may be partially explained by changes in growth rate (Supplementary Note 2 and Supplementary Fig. 14), other phenomena such as off-target binding and stress-responses to the TAs may be at play (see Supplementary Note 3). Detailed investigation of how these specific promoters respond to stress inflicted by TAs is another avenue of future research that can contribute to the predictive power of the resource sharing model for specific promoters.

To decrease the knockdown in a GOI's expression level due to expression of a TA, the expression of the TA could be limited to the minimal level that provides the desired TA-driven output activation[61] (Supplementary Fig. 2c, d). While reducing the concentration of a resource competitor is a viable approach to reduce loading effects, expressing sufficient amounts of TA to maximize TA-driven gene expression will typically lead to substantial resource loading and hence to knockdown of non-target genes (Supplementary Fig. 2d). Our iFFL decouples GOI expression from the levels of TAs, thereby eliminating the need to simultaneously optimize the expression of both target and non-target TA genes by scanning levels of the TA. Instead, the TA can be set to any desired level to achieve a given amount of TA-driven output expression without consequence for non-target feedforward-regulated GOIs.

While methods to make a GOI's expression level robust to the variability of gene expression resources have been demonstrated in bacterial cells using feedback control[71–73], no previous reports have solved this problem in eukaryotic cells. Our feedforward controller and a miRNA-based device described in a concurrent report[74] represent the first eukaryotic solutions to decouple expression of a GOI from variable gene expression resources. Among the cited existing bacterial solutions, only the sRNA-based feedback controller developed by Huang et al. in *E. coli*[72] has achieved near-perfect adaptation to resource loading, similar to our feedforward controller in mammalian cells. However, the bacterial solution of Huang et al.[72] is not transportable to mammalian cells because it uses prokaryotic-specific parts (sRNA and sigma factors) and is designed to adapt to loads in translational but not transcriptional resources, which are the major contributors to resource variability in mammalian cells (Supplementary Fig. 4). Compared to miRNA-based implementations, such as in the concurrent report[74], both miRNA-based and endoRNase-based iFFL can mitigate effects of resource loading on the expression level of a GOI. However, due to translational amplification, endoRNase-based iFFLs benefit from a higher production rate of the effector molecule. This contributes to smaller a feedforward impedance $\epsilon$ and thus enhanced robustness of the endoRNase-based iFFLs (see Supplementary Note 5 and Supplementary Fig. 40). Furthermore, expression of endoRNases but not of miRNAs directly requires translational resources. As a consequence, iFFLs utilizing endoRNases can, in principle, also adapt to changes in the availability of translational resources (see Supplementary Note 5).

In summary, the performance of genetic devices across various cell types and changing cellular conditions is greatly affected by the cellular environment, and in large part, depends on the available gene expression resources. The availability of these resources, in turn, becomes highly variable when gene expression

changes during a circuit's operation. The endoRNase-based feedforward controller provides a readily-usable solution to maintain robust gene expression, despite variable levels of resources. Since the controller is highly portable, it can be easily implemented to enable robust control of gene expression across a number of mammalian synthetic biology applications, such as cell-based therapies, gene therapies, and organoids. More generally, the endoRNase-based feedforward controller enables predictable modular composition of engineered genetic systems in mammalian cells and can function as a general-purpose tool for the design of sophisticated systems that perform as predicted across variable contexts.

## Methods

**A mathematical model to guide iFFL design.** We first define the TX marker vs output dose–response curve. The steady-state concentration of $z$, the TX marker, can be written as $z = V_z \cdot D \cdot R$, where $V_z$ is a lumped parameter independent of $D$ and $R$, and defined similarly to $V_y$ in Eq. (2) (see Supplementary Note 5 for details). Substituting $D \cdot R = z/V_z$ into Eq. (1), we obtain the output level as a function of the experimentally measurable quantity $z$:

$$y = V_y \cdot \frac{z/V_z}{1 + z/(V_z \epsilon)}. \tag{3}$$

This TX marker vs output dose–response curve is shown in Fig. 3b. Its shape can be characterized by two metrics $Z_{50}$ and $Y_{max}$. Specifically, as $z \to \infty$, $Y_{max} = V_y \epsilon$. $Z_{50}$ is the TX marker's fluorescence level at which the iFFL module's output is half of $Y_{max}$, which can be computed as $Z_{50} := V_z \epsilon$ by Eq. (3).

We next quantified the feedforward impedance $\epsilon$ for iFFL modules with different numbers of uORFs in the 5′ UTR of the CasE transcription unit. With reference to Fig. 3c, the relationship between $n$ and $\kappa_x$ has been experimentally characterized in Gam et al.[44], where the authors measured the expression of a constitutive fluorescent protein p with different numbers of uORFs in the 5′ UTR of its transcript. Since the expression level of a constitutive gene is inversely proportional to the dissociation constant between ribosomes and its transcript (i.e., $p \propto 1/\kappa_x$, see Supplementary Note 5), we have

$$\text{relative } \kappa_x = (\text{ relative } \kappa_x)(n) := \frac{\kappa_x(n)}{\kappa_x(0)} = \frac{p(0)}{p(n)}, \tag{4}$$

where $p(n)$ and $\kappa_x(n)$ are the steady-state expressions of p and the dissociation constant between ribosomes and protein p's mRNA transcript in the presence of $n$ uORFs, respectively. Since we have derived from Eq. (3) that (i) $Y_{max}$ and $Z_{50}$ both are proportional to $\epsilon$ and hence proportional to $\kappa_x$ and that (ii) $\kappa_x(n) = (\text{relative } \kappa_x)(n) \times \kappa_x(0)$ according to Eq. (4), our model predicts that $Y_{max} = Y_{max}(n)$ and $Z_{50} = Z_{50}(n)$ are both proportional to relative $\kappa_x$.

In addition to robustness to variation in free transcriptional and translational resource concentrations, the iFFL can also attenuate the effect of DNA copy number variation (i.e., changes in $D$) on the module's output. Since $D$ and $R$ are clustered together in Eq. (1), our analysis on the module's robustness to $R$ carries over directly when analyzing its robustness to $D$: when $DR \gg \epsilon$, we have $y \approx V_y \epsilon$ according to Eq. (1), which is independent of $D$. Robustness to variations in $D$ also includes temporal variability of DNA concentration, which is present in transient transfection experiments due to dilution of DNA plasmids as cells grow and divide. As one decreases the number of uORFs in the endoRNase's transcript, our model predicts that the iFFL module becomes more robust to DNA copy number variability in the sense that it's output remains the same for a wider range of DNA copy numbers (i.e., smaller $Z_{50}$). This allows the module's output to maintain $Y_{max}$ for a longer period of time as DNA concentration gradually decreases, a phenomenon we observed both experimentally (see Fig. 6d and Supplementary Fig. 39) and numerically (see Supplementary Fig. 40).

**Modular plasmid-cloning scheme.** Plasmids were constructed using a modular Golden Gate strategy similar to previous work in our lab[44,69]. Briefly, basic parts (insulators, promoters, 5′ UTRs, coding sequences, 3′ UTRs, and terminators—termed level 0s (pL0s)) were created via standard cloning techniques. Typically, pL0s were generated via PCR (Q5 or OneTaq hot-start polymerases, New England BioLabs (NEB)) followed by In-Fusion (Takara Bio) or digestion/ligation with the pL0 backbones; in addition, we also utilized direct synthesis of shorter inserts followed by ligation with T4 ligase into pL0 backbones. Oligonucleotides were synthesized by Integrated DNA Technologies (IDT) or SGI-DNA. pL0s were assembled into transcription units (TUs—termed level 1s (pL1s)) using BsaI Golden Gate reactions (10–20 cycles between 16 °C and 37 °C, T4 ligase). TUs were assembled into multi-TU plasmids using SapI Golden Gate reactions. To make lentivirus transfer plasmids, pL0s or pL1s were cloned into a new vector (pLV-RJ v4F) derived from pFUGW (AddGene plasmid #14883) using either BsaI or SapI Golden Gate, respectively. All restriction enzymes and T4 ligase were obtained from NEB. Plasmids were transformed into Stellar *E. coli* competent cells (Takara Bio). Transformed Stellar cells were plated on LB agar (VWR) and

propagated in TB media (Sigma-Aldrich). Carbenicillin (100 μg/mL), kanamycin (50 μg/mL), and/or spectinomycin (100 μg/mL) were added to the plates or media in accordance with the resistance gene(s) on each plasmid. All plasmids were extracted from cells with QIAprep Spin Miniprep and QIAGEN Plasmid Plus Midiprep Kits. Plasmid sequences were verified by Sanger sequencing at Quintara Biosciences. Genbank files for each plasmid and vector backbone used in this study, as well as primers and cloning details, are provided in Source Data. Plasmid sequences were created and annotated using Geneious (Biomatters). New plasmids used in this study are available on Addgene (http://www.addgene.org/Ron_Weiss/).

**Estimation of CpG island size in plasmids.** The size of CpG islands in constitutive promoters (see Supplementary Fig. 12) were estimated using the CpG Islands v1.1 tool in Geneious (Thobias Thierer & Biomatters). The number of bases classified as part of a CpG island (not necessarily contiguous) were summed and presented in the figure. Plasmid maps are annotated with the highest-confidence bases of the CpG islands.

**Cell culture.** HEK-293 cells (ATCC), HEK-293FT cells (Thermo Fisher), HeLa cells (ATCC), and Vero 2.2 cells (Massachusetts General Hospital) were maintained in Dulbecco's modified Eagle media (DMEM) containing 4.5 g/L glucose, L-glutamine, and sodium pyruvate (Corning) supplemented with 10% fetal bovine serum (FBS, from VWR). CHO-K1 cells (ATCC) were grown in F12-K media containing 2 mM L-glutamine and 1500 mn/L sodium bicarbonate (ATCC) supplemented with 10% FBS. U2OS cells (ATCC) were grown in McCoy's 5A media with high glucose, L-glutamine, and bacto-peptone (Gibco) supplemented with 10% FBS. All cell lines used in the study were grown in a humidified incubator at 37 °C and 5% $CO_2$. All cell lines tested negative for mycoplasma.

**Transfections.** Cells were cultured to 90% confluency on the day of transfection, trypsinized, and added to new plates simultaneously with the addition of plasmid-transfection reagent mixtures (reverse transfection). Transfections were performed in 24-well or 96-well pre-treated tissue culture plates (Costar). Following are the volumes, number of cells, and concentrations of reagents used for 96-well transfections; for 24-well transfections, all values were scaled up by a factor of 5. In total, 120 ng of total DNA was diluted into 10 μL Opti-MEM (Gibco) and lightly vortexed. The transfection reagent was then added, and samples were lightly vortexed again. The DNA–reagent mixtures were incubated for 10–30 min, while cells were trypsinized and counted. After depositing the transfection mixtures into appropriate wells, 40,000 HEK-293, 40,000 HEK-293FT, 10,000 HeLa, 20,000 CHO-K1, 20,000 Vero 2.2, or 10,000 U2OS cells suspended in 100 μL of media were added. The reagent used in each experiment along with plasmid quantities per sample and other experimental details are provided in Source Data. Lipofectamine LTX (ThermoFischer) was used at a ratio of 1 μL of PLUS reagent and 4 μL of LTX per 1 μg of DNA. PEI MAX (Polysciences VWR) was used at a ratio of 3 μL of PEI per 1 μg of DNA. Viafect (Promega) was used at a ratio of 3 μL Viafect per 1 μg of DNA. Lipofectamine 3000 was used at a ratio of 2 μL P3000 and 2 μL Lipo 300 per 1 μg of DNA. Attractene (Qiagen) was used at a ratio of 5 μL of attractene per 1 μg of DNA. For experiments with measurement windows between 12–72 h (as indicated on the figures or in their captions), the media of the transfected cells was not replaced between transfection and data collection. For experiments with measurements at longer time points, the transfected cells were passaged at 72 h in fresh media on a new plate. In order to maintain a similar number of cells for data collection at longer time points, transfected cells were split at ratios of 1:2 or 1:4 for samples being collected at 96 or 120 h, respectively. For all transfections with doxycycline (Dox, Sigma-Aldrich), Dox was added immediately after transfection; an exception is an experiment shown in Supplementary Fig. 9, in which Dox was added 24 h after transfection.

In each transfection sample, we included a hEF1a-driven transfection marker to indicate the dosage of DNA delivered to each cell and to facilitate consistent gating of transfected cells. Of the strong promoters we tested (CMV, CMVi, and hEF1a), the hEF1a promoter gave the most consistent expression across cell lines and was generally less affected by resource loading by Gal4 TAs (Supplementary Figs. 1, 11, 12, and 16). The data in Supplementary Fig. 20 used CMV promoters for all transcription units (including the transfection marker).

**Lentivirus production and infection.** Lentivirus production was performed using HEK-293FT cells and second-generation helper plasmids MD2.G (Addgene plasmid #12259) and psPax2 (Addgene plasmid #12260). HEK-293FT cells were grown to 90% confluency, trypsinized, and added to new pre-treated 10-cm tissue culture plates (Falcon) simultaneously with the addition of plasmid-transfection reagent mixtures. Four hours before transfection, the media on the HEK-293FT cells was replaced. To make the mixtures, first 3 μg psPax2, 3 μg pMD2.g, and 6 μg of the transfer vector were diluted into 600 μL Opti-MEM and lightly vortexed. In total, 72 μL of FuGENE6 (Promega) was then added, and the solution was lightly vortexed again. The DNA-FuGENE mixtures were incubated for 30 min, while cells were trypsinized and counted. After depositing the transfection mixtures into appropriate plates, $6 \times 10^6$ HEK-293FT cells suspended in 10 mL of media were added. Sixteen hours after transfection, the media was replaced. Forty-eight hours

after transfection, the supernatant was collected and filtered through a 0.45 PES filter (VWR).

For infections, HEK-293FT cells were grown to 90% confluency, trypsinized, and $1 \times 10^6$ cells were resuspended in 1 mL of media. The cell suspension was mixed with 1 mL of viral supernatant, then the mixture was added to a pre-treated six-well tissue culture plate (Costar). To facilitate viral uptake, polybrene (Millipore-Sigma) was added to a final concentration of 8 μg/mL. Cells infected by lentiviruses were expanded, and cultured for at least 2 weeks before use in experiments using the same conditions for culturing HEK-293FT cells as described above.

**RT-qPCR.** Transfections for qPCR were conducted in 24-well plates (Costar). RNA was collected 48 h after transfection with the RNeasy Mini kit (Qiagen). Reverse transcription was performed using the Superscript III kit (Invitrogen) following the manufacturer's recommendations. Real-time qPCR was performed using the KAPA SYBR FAST qPCR 2X master mix (Kapa Biosystems) on a Mastercycler ep Realplex (Eppendorf) following the manufacturer's recommended protocol. Primers for the CMV-driven output (mKate) targeted the coding sequence. Primers for 18S rRNA were used as an internal control for normalization. The qPCR calculations are provided in Source Data.
Primers:
mKate (CMV:output) forward: GGTGTCTAAGGGCGAAGAGC
mKate (CMV:output) reverse: GCTGGTAGCCAGGATGTCGA
18S forward: GTAACCCGTTGAACCCCATT
18S reverse: CCATCCAATCGGTAGTAGCG.

**Flow cytometry.** To prepare samples in 96-well plates for flow cytometry, the following procedure was followed: media was aspirated, 50 μL PBS (Corning) was added to wash the cells and remove FBS, the PBS was aspirated, and 40 μL Trypsin-EDTA (Corning) was added. The cells were incubated for 5–10 min at 37 °C to allow for detachment and separation. Following incubation, 80 μL of DMEM without phenol red (Gibco) with 10% FBS was added to inactivate the trypsin. Cells were thoroughly mixed to separate and suspend individual cells. The plate(s) were then spun down at $400 \times g$ for 4 min, and the leftover media was aspirated. Cells were resuspended in 170 μL flow buffer (PBS supplemented with 1% BSA (Thermo Fisher), 5 mM EDTA (VWR), and 0.1% sodium azide (Sigma-Aldrich) to prevent clumping). For prepping plates of cells with larger surface areas, all volumes were scaled up in proportion to surface area and samples were transferred to 5-mL polystyrene FACS tubes (Falcon) after trypsinization. For standard co-transfections, 10,000–50,000 cells were collected per sample. For the poly-transfection experiment and transfections into cells harboring an existing lentiviral integration, 100,000–200,000 cells were collected per sample.

For the experiments shown in Fig. 1 and Supplementary Figs. 2, 3, 6, and 8, samples were collected on a BD LSR II cytometer located in the Koch Institute Flow Cytometry Core equipped with a 405-nm laser with 450/50-nm filter ("Pacific Blue") for measuring TagBFP or EBFP2, 488 laser with 515/20 filter ("FITC") for measuring EYFP or mNeonGreen, 561-nm laser with 582/42-nm filter ("PE") or 610/20-nm filter ("PE-Texas Red") for measuring mKate2 or mKO2, and 640 laser with 780/60-nm filter ("APC-Cy7") for measuring iRFP720. For all other experiments, samples were collected on a BD LSR Fortessa located in the MIT Synthetic Biology Center equipped with a 405-nm laser with 450/50-nm filter ("Pacific Blue") for measuring TagBFP or EBFP2, 488 laser with 530/30 filter ("FITC") for measuring EYFP or mNeonGreen, 561-nm laser with 582/15-nm filter ("PE") or 610/20-nm filter ("PE-Texas Red") for measuring mKate2 or mKO2, and 640 laser with 780/60-nm filter ("APC-Cy7") for measuring iRFP720. In all, 500–2000 events/s were collected either in tubes via the collection port or in 96-well plates via the high-throughput sampler (HTS). All events were recorded, and compensation was not applied until processing the data (see below).

**Intracellular antibody staining.** HA-tagged Gal4 TAs were stained with anti-HA.11 directly conjugated to Alexa Fluor 594 (BioLegend catalogue #901511, clone 16B12, isotype IgG1 κ). As a control for non-specific anti-HA binding, untransfected cells were stained with the same antibody. Cellular Ki-67 was stained with anti-Ki-67 directly conjugated to PE/Dazzle 594 (BioLegend catalogue #350533, isotype IgG1 κ). As a control for non-specific anti-Ki-67 binding, cells were stained with an IgG1 κ isotype control directly conjugated to PE/Dazzle 594 (BioLegend catalogue #400177).

Staining was performed on cells grown in 96-well plates. Cells were washed with PBS, trypsinized, and separated into individual cells as described above for preparing samples for flow cytometry. After quenching the trypsin reaction and mixing into a single-cell suspension, cells were transferred to U-bottom plates and pelleted. All centrifugation steps with plates occurred at $400 \times g$ for 4 min. After pelleting, the media–trypsin mix was aspirated, and the cells were fixed via incubation in 50 μL of 4% formaldehyde (BioLegend) for 20 min at room temperature. After fixation, the cells were pelleted, the fixation buffer was removed, and the cells were resuspended in 50 μL Intracellular Staining Permeabilization Wash Buffer (BioLegend). Antibodies were added to each well using the manufacturer's recommended volumes; then plates were placed on a nutator in the dark in a cold room (4 °C) overnight. After incubation with the antibody, the cells

were washed three times with 50 μL permeabilization buffer, then resuspended in 170 μL of flow buffer (see above for formulation).

**Flow-cytometry data analysis**. Analysis of flow-cytometry data was performed using our MATLAB-based flow-cytometry analysis pipeline (https://github.com/Weiss-Lab/MATLAB_Flow_Analysis). Basic processing steps with example data are shown in Supplementary Fig. 41. Briefly, single cells were isolated by drawing morphological gates based on cellular side-scatter and forward-scatter. Arbitrary fluorescence units were converted to standardized molecules of equivalent fluorescein (MEFL) units using RCP-30-5A beads (Spherotech) and the TASBE pipeline process[75]. Fluorescence compensation was performed by subtracting auto-fluorescence (computed from wild-type cells), computing linear fits between channels in single-color transfected cells, then using the fit slopes as matrix coefficients for matrix-based signal de-convolution. Threshold gates were manually drawn for each channel based on the fluorescence of untransfected cells. Generally, transfected cells within a sample were identified by selecting cells that pass either the gate for the output of interest (output[+]) or the gate for the transfection marker (TX marker[+]). Binning was performed by defining bin edges, then sorting cells into a bin if expression of the reporter used for binning was less than or equal to the high-bin edge and greater than the low-bin edge. Median fluorescence levels were used for summary statistics so as not to make any assumptions about the expression distributions. In order to avoid the artefact of negative fold changes, non-positive fluorescence values were discarded prior to making measurements on binned or gated populations.

The density of cells in scatterplots was estimated by sorting the cells into 25 evenly spaced bins in each dimension (for $N$ dimensions, $25^N$ total bins), finding the number of cells in each bin, then linearly interpolating the density for each individual cell using the bin centers as the interpolation nodes. Density was calculated with either the log- or biexponentially-transformed data (see plot axes) because the dominant variance is approximately log-distributed. The outer boundaries of the bins in each dimension were automatically found by taking the minimum and maximum values of the data, then respectively subtracting and adding 5% of the log/biexponential range between min and max.

In Fig. 2, our library of constitutive promoters had different nominal expression levels and were variably affected by resource loading. We thus include a discussion and examples of how fluorescent gating strategies affect the measurements of expression and fold changes in Supplementary Note 1 and Supplementary Fig. 42. Some promoters drove expression that was nearly undetectable (Supplementary Fig. 43). In order to limit the bias in our reporting of minimally affected promoters by the proximity of {P}:output$_1$ expression to autofluorescence, our analysis of this data incorporates an additional autofluorescence subtraction step described in Supplementary Note 1. A comparison of the differences in fold changes with and without this additional autofluorescence subtraction is shown in Supplementary Fig. 44a. This step reduced the correlation between the nominal output levels of {P}:output$_1$ and the fold changes in response to resource loading by Gal4 TAs (Supplementary Fig. 44b).

When first analyzing the data in Fig. 4, we found that the measurements of fold changes and robustness for the UR variants with diluted output plasmid DNA were sensitive to the fluorescent gating strategy used in the analysis. Our typical gating routine of selecting cells positive for either the output or the transfection marker yielded fold changes of the diluted UR variants that were much larger than when gating on cells positive for just the output. Conversely, both gating strategies yielded similar fold changes for the iFFL variants regardless of their nominal output. We suspect that the difference in measurements for the diluted UR variants may result from (i) reduced UR plasmid uptake when forming lipid–DNA complexes for co-transfection with the Gal4-VPR plasmid (which is larger than the DNA-mass-offsetting plasmid Gal4-None) and/or (ii) repression of UR output expression below the autofluorescence threshold. Since these confounding factors could not be distinguished, we report the results for the cells gated positive for just the output (which more conservatively estimates fold changes in the output of the UR system) in the main figures and include results for gating cells positive for either the output or the transfection marker in Supplementary Figs. 19 and 23 for comparison. For the hEF1a iFFL, we also include comparisons of results with both gating strategies in Supplementary Figs. 25–28.

**Calculation of fold changes and robustness scores**. For quantifying the effects of resource loading, we measured fold changes by dividing the median output level of each sample by that of the equivalent sample in the absence of resource loading (i.e., the nominal output level of the module of interest). The nominal output is defined as the level of output in the presence of either Gal4-None (Gal4 DBD only, used directly when comparing Gal4 TAs) or 0 ng Gal4-{AD} (used in dose-responses).

$$\text{Fold-}\Delta(\text{Gal4-}\{AD\}) = \frac{\text{Output}(\text{Gal4-}\{AD\})}{\text{Output}(\text{Gal4-None})} \quad (5)$$

$$\text{Fold-}\Delta(\text{Gal4-}\{AD\} = x) = \frac{\text{Output}(\text{Gal4-}\{AD\} = x)}{\text{Output}(\text{Gal4-}\{AD\} = 0)}, \quad (6)$$

where $log_2$-transformed fold changes are shown for experiments with multiple repeats, the values shown are the mean of the $log_2$-transformed fold changes, rather than the $log_2$-transformation of the mean of the fold changes. This order of operations ensures that standard deviations of the fold changes can be computed directly on the $log_2$-transformed scale.

For comparing UR and iFFL variants, we also computed robustness scores from the fold changes using the formulae below:

$$\text{Robustness}(\text{Gal4-}\{AD\}) = 100 \cdot (1 - |1 - \text{Fold-}\Delta(\text{Gal4-None})|) \quad (7)$$

$$\text{Robustness}(\text{Gal4-}\{AD\} = x) = 100 \cdot (1 - |1 - \text{Fold-}\Delta(\text{Gal4-}\{AD\} = x)|). \quad (8)$$

**Estimation of cell density by flow cytometry**. As a post-hoc method of measuring the effects of toxicity in samples transfected with resource competitors, we estimated the cell density observed at the time of fluorescence measurements. When collecting flow-cytometry data, we typically constrained the number of events collected, making the count of cells per sample not representative of the total number of cells per well. We thus instead estimated cell density in a given sample with the following formula:

$$[\text{Cells}] \ (\text{cells} \cdot \mu L^{-1}) = \frac{\text{Eventrate} \ (\text{cells} \cdot s^{-1})}{\text{Flowrate} \ (\mu L \cdot s^{-1})}.$$

To compute the event rate, we estimated the number of cells (i.e., events passing morphological gating) collected per second in each sample. The length of time between the measurements of individual cells in flow cytometry approximately follows an exponential distribution. We thus fit an exponential distribution using the MATLAB function "fitdist()" (https://www.mathworks.com/help/stats/fitdist.html) to the differences between time-stamps of sequentially collected cells. Before fitting, we removed inter-cell times larger than the 99.9th percentile to prevent biasing by large outliers. The characteristic parameter of the exponential distribution $\lambda$ is the inverse of the average time between events. Thus, the event rate is given by $\frac{1}{\lambda}$, which is also the mean of the exponential distribution.

To ensure a known and controlled flow rate, any samples with concentration measured were collected via the HTS attached to the flow cytometer. The flow rate of the HTS can be set through the FACSDiva Software (BD) controlling the instrument. The flow rate of each sample was recorded and input into the calculation. Because changes in the overall density of cells in a sample depends both on the potency of growth inhibition by transfected genes as well on the fraction of cells transfected, we only analyzed in-depth and reported values for samples from HEK cell lines. The other cell lines (HeLa, CHO-K1, Vero 2.2, and U2OS) were generally too poorly transfected to achieve reliable and sensitive measurements of changes in cell density as a function of transfected toxic genes.

**Model fitting**. Where possible, fluorescent reporters were used to estimate the concentration of a molecular species for the purpose of model fitting. For fitting the Gal4 TA dose–response curves (both on-target activation and off-target resource loading) in Fig. 1 and Supplementary Fig. 2, we used a fluorescent marker co-titrated with the Gal4 activators (Gal4 marker) to approximate the amount of Gal4 delivered per cell. The Gal4 marker correlated with the DNA dosage with an $R^2$ value of 0.86 or better for each experimental repeat (Supplementary Fig. 3a). However, the sensitivity of activation to Gal4 levels made the measurements as a function of Gal4 DNA dosage relatively noisy between experimental repeats (Supplementary Fig. 3b–e). Thus, the marker levels could more accurately estimate the amount of Gal4 expressed in the median cell than the DNA dosages.

For fitting both the resource sharing and iFFL models, we used the MATLAB function "lsqcurvefit()" (https://www.mathworks.com/help/optim/ug/lsqcurvefit.html), which minimizes the sum of the squares of the residuals between the model and the data. As the function input values, we used the level of either the Gal4 TA (in the case of resource sharing—as measured by Gal4 marker) or the transfection marker (in the case of the iFFL). For fitting the Gal4 TA dose–response data, the residuals were computed between the median CMV:output$_1$ or UAS:output$_2$ levels and function outputs directly. In addition, all median values computed from different experimental repeats were pooled together before fitting. For fitting iFFL and UR models, the residuals were computed between the $log_{10}$- and biexponentially-transformed levels of the output protein of interest and the $log_{10}$- and biexponentially-transformed function outputs, respectively. In experiments with the hEF1a iFFL being tested only in HEK-293FT cells, the entire morphologically gated population of cells was used for fitting. In hEF1a iFFL experiments containing multiple cell types, to prevent the model from over-fitting the untransfected population in more difficult-to-transfect cells, the cells in each sample were analytically binned into half-log-decade-width bins based on the transfection marker, and an equivalent number of cells from each bin were extracted, combined, and used for fitting. In samples with the CMVi iFFL, the relatively high expression of the CMVi promoter compared to the hEF1a promoter (which is used as a transfection marker and proxy for DNA/resource input level $z$) in most cell lines imposes nonlinearity in the transfection marker vs output curve at low plasmid DNA copy numbers per cell. This nonlinearity led us to gate cells positive for either the iFFL output or the transfection marker for fitting. For the resource sharing models, all parameters for all Gal4 TAs were fit simultaneously using a custom function, "lsqmultifit()", that was created based on "nlinmultifit()" on the MATLAB file exchange (https://www.mathworks.com/matlabcentral/fileexchange/40613-multiple-curve-fitting-with-common-parameters-using-nlinfit).

The goodness of fit was measured by computing the normalized root-mean-square error CV(RMSE) using the following formula:

$$\mathrm{CV}\,(\mathrm{RMSE}) = \frac{\sqrt{\frac{1}{\bar{y}}\sum_i (y(x_i) - f(x_i))^2}}{\bar{y}},$$

where $y(x_i)$ is the value of the data at the input value $x_i$, $\bar{y}$ is the mean of $y$ for all values of $x$, and $f(x_i)$ is the function output at input value $x_i$.

Resource loading characterization data were fit with the following equations (see Supplementary Note 3 for more details):

$$\mathrm{Output}_1 = \alpha_1 \cdot \frac{\frac{R_{\mathrm{TX}}}{k_{12}}}{1 + \frac{R_{\mathrm{TX}}}{k_{12}}}, \tag{9}$$

$$\mathrm{Output}_2 = \alpha_2 \cdot \frac{\frac{R_{\mathrm{TX}}}{k_{22}} \cdot \left(\frac{u_2}{k_{21}}\right)^2}{1 + \left(\frac{u_2}{k_{21}}\right)^2 \cdot \left(1 + \frac{R_{\mathrm{TX}}}{k_{22}}\right)}, \tag{10}$$

iFFL data were fit with Eq. (3) above, which is reproduced here for convenience:

$$y = V_y \cdot \frac{z/V_z}{1 + z/(V_z\epsilon)}.$$

For other comparisons where we present values of $r$ or $R^2$, the former is the Pearson's correlation computed with the MATLAB function "regression()" (https://www.mathworks.com/help/deeplearning/ref/regression.html), and the latter is the coefficient of determination between predicted and actual values.

**Reporting summary**. Further information on research design is available in the Nature Research Reporting Summary linked to this article.

## Data availability

New plasmids used in this study are available for distribution from Addgene (http://www.addgene.org/Ron_Weiss/). Raw .fcs files are available from the corresponding authors upon reasonable request. Source data are provided with this paper.

## Code availability

General MATLAB code for use in .fcs file processing and analysis are available under an open-source license in our GitHub repository at https://github.com/Weiss-Lab/MATLAB_Flow_Analysis. Specific .m scripts for each experiment are available from the corresponding authors upon reasonable request.

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

## Acknowledgements

We would like to acknowledge Douglas Lauffenburger and Ahmad Khalil for helpful discussion, Jeremy Gam and Katherine Kiwimagi for plasmids, and Samira Kiani for initially identifying potential resource sharing problems in mammalian cells while in Prof. Weiss' research group. The hMDM2 promoter (MDM2p-Mdm2-YFP) was a gift from Uri Alon and Galit Lahav (Addgene plasmid #53962). MD2.G and psPax2 were gifts from Didier Trono (Addgene plasmid #12259 & #12260). pFUGW was a gift from David Baltimore (Addgene plasmid #14883). Vero 2.2 cells were a gift from Xandra Breakefield. This work was supported by the National Institutes of Health (NIH P50GM098792), the National Science Foundation (MCB-1840257), and the United States Air Force Office of Scientific Research (FA9550-14-1-0060).

## Author contributions

R.D.J., Y.Q., V.S., R.W., and D.D.V. designed the study; R.D.J., V.S., B.D., and J.H. performed the experiments; R.D.J., V.S., and B.D. analyzed the data; Y.Q. and R.D.J. developed the mathematical models; R.D.J., Y.Q., R.W., and D.D.V. wrote the paper.

## Competing interests

The Massachusetts Institute of Technology has filed a patent application on behalf the inventors (R.D.J., Y.Q., B.D., R.W., and D.D.V.) of the endoRNase-based iFFL design described (US Provisional Application No. 62/992,82). The remaining authors declare no competing interests.
