## [Peer Review File · Nature Communications]

Reviewers' Comments:

Reviewer #1:

Remarks to the Author:

The goal of this manuscript is to investigate how resource loading affects the function of mammalian synthetic gene circuits and to develop incoherent feedforward loop (iFFL)-based strategies that counteract such effects. The experimental model system used to investigate these effects consists of two modules. The first module is a CMV promoter-driven constitutive "Output" reporter gene, which detects the effects of loading by the second component: a second reporter activated by a transcription factor binding to 5 UAS elements in the modified hEF1alpha promoter of a second reporter. The transcriptional activators have a Gal4 DNA binding domain and various activator domains (VP64, VP16, Rta, p65, VPR). As the amount of Gal4 marker (a fluorescent marker co-titrated with the Gal4 module) increases, the level of Output protein drops. The same effect is observed for many constitutive promoters and several widely used mammalian cell lines. Next, co-expressing Output with the endoribonuclease CasE that targets Output's 3' end implements an iFFL that provides adaptation to resource loading, although at the expense of lowering maximum Output expression. These effects are preserved across various cell lines. Moreover, CasE-based iFFL also provides robustness to DNA copy number variation. A detailed mathematical model capturing most aspects of the data strengthens the conclusions.

I have reviewed this manuscript for another journal, and I am impressed by how the Authors have addressed in their revision not only my comments, but also the comments of the other Reviewer. The study is highly interesting, and the experimental methods are elegant. Testing across activators and cell lines, including integrated constructs ensure relevance to many mammalian synthetic biology labs and beyond. The strong mathematical modeling component supports the conclusions. The research is well executed, and the manuscript text and figures convey the message well. Importantly, the limitations of the iFFL system are also well articulated. Overall, I would like to recommend strongly this interesting and well-executed study for publication in Nature Communication without any further changes.

Reviewer #2:

Remarks to the Author:

Thank you for addressing my comments.

Reviewer #3:

Remarks to the Author:

With the revision, the authors have very adequately addressed all points raised in the first round of review. In particular, extended analysis (on trade-offs induced by growth effects and on the direct comparison of the performance of CasE- and miRNA-based incoherent feedforward loop designs for adaptation to resource loading) substantially strengthens the manuscript. Clarifications regarding context, related work, and major claims have the same effect. Overall, as stated before, the study

reaches its potential of providing important advances for the quantitative analysis of resource competition in mammalian cells, with broad implications for the design of novel synthetic circuits.

Minor comments:

(i) The finding of more effective targeting for both CasE and miRNA via 5'UTR than via 3'UTR is notable for future designs. An explanation for this effect (in terms of previous observation of targeting efficiency; l. 158) exists – could you speculate why it is observed for both molecular implementations?

(ii) SI, l. 1121: ‘... suggesting that the effect of the Gal4 TAs on its transcription and the [the] growth rate ...’

Author responses are below in blue text

REVIEWERS' COMMENTS:

Reviewer #1 (Remarks to the Author):

The goal of this manuscript is to investigate how resource loading affects the function of mammalian synthetic gene circuits and to develop incoherent feedforward loop (iFFL)-based strategies that counteract such effects. The experimental model system used to investigate these effects consists of two modules. The first module is a CMV promoter-driven constitutive "Output" reporter gene, which detects the effects of loading by the second component: a second reporter activated by a transcription factor binding to 5 UAS elements in the modified hEF1alpha promoter of a second reporter. The transcriptional activators have a Gal4 DNA binding domain and various activator domains (VP64, VP16, Rta, p65, VPR). As the amount of Gal4 marker (a fluorescent marker co-titrated with the Gal4 module) increases, the level of Output protein drops. The same effect is observed for many constitutive promoters and several widely used mammalian cell lines. Next, co-expressing Output with the endoribonuclease CasE that targets Output's 3' end implements an iFFL that provides adaptation to resource loading, although at the expense of lowering maximum Output expression. These effects are preserved across various cell lines. Moreover, CasE-based iFFL also provides robustness to DNA copy number variation. A detailed mathematical model capturing most aspects of the data strengthens the conclusions.

I have reviewed this manuscript for another journal, and I am impressed by how the Authors have addressed in their revision not only my comments, but also the comments of the other Reviewer. The study is highly interesting, and the experimental methods are elegant. Testing across activators and cell lines, including integrated constructs ensure relevance to many mammalian synthetic biology labs and beyond. The strong mathematical modeling component supports the conclusions. The research is well executed, and the manuscript text and figures convey the message well. Importantly, the limitations of the iFFL system are also well articulated. Overall, I would like to recommend strongly this interesting and well-executed study for publication in Nature Communication without any further changes.

Thank you!

Reviewer #2 (Remarks to the Author):

With the revision, the authors have very adequately addressed all points raised in the first round of review. In particular, extended analysis (on trade-offs induced by growth effects and on the direct comparison of the performance of CasE- and miRNA-based incoherent feedforward loop designs for adaptation to resource loading) substantially strengthens the manuscript. Clarifications regarding context, related work, and major claims have the same effect. Overall, as stated before, the study reaches its potential of providing important

advances for the quantitative analysis of resource competition in mammalian cells, with broad implications for the design of novel synthetic circuits.

Minor comments:

(i) The finding of more effective targeting for both CasE and miRNA via 5'UTR than via 3'UTR is notable for future designs. An explanation for this effect (in terms of previous observation of targeting efficiency; l. 158) exists – could you speculate why it is observed for both molecular implementations?

Thank you for bringing this up, as we should have addressed it better in the manuscript - we think that this has to do with imperfect suppression of translation by 3' cleavage. For the models, we assume that cleavage by the endoRNase or miRNA eliminates the mRNA as a species that can productively produce protein. However, while cleavage of the 5' UTR de-caps the mRNA (which is detrimental to translation initiation), cleavage of the 3' UTR leaves an intact coding sequence that can be translated (though with much-reduced stability and likely also reduced translation efficiency due to de-circularization). Considering specifically Cas6-family endoRNases like CasE, they bind to an RNA hairpin and cleave at the 3' base of its stem. After cleavage, they retain high affinity for the hairpin, which is on the 5' side of the cleavage site. For a target site in the 3' UTR, this means the endoRNase remains bound to the 3' terminus of the main transcript sequence. Previous work has shown that this property can be used to stabilize RNAs that lack a poly-A tail¹, indicating that the endoRNase provides some degree of stabilization against exonucleases while bound to RNA. Conversely, the RISC complex retains some affinity for the RNA sequence to the 3' side of their cleavage site², indicating that this protective mechanism is probably not conserved for both implementations, and thus is not required for the observed behavior, though it may contribute to it. We have added a few sentences to the Discussion to summarize these points:

Our iFFL models assume that the output mRNA species is completely destroyed when cleaved by an endoRNase/miRNA. However, whereas 5' cleavage removes the 5' cap, which is detrimental to translation initiation³, 3' cleavage may leave the transcript competent for continued translation. In addition, Cas6-family endoRNases like CasE can remain tightly bound to the sequence of RNA to the 5' side of their cleavage site and protect the bound strand from 3' exonucleases¹. However, this protective mechanism is not likely to be the sole cause for the observed differences, as for miRNAs, the RISC complex instead retains moderate affinity for the sequence to the 3' side of its cleavage site².

(ii) SI, l. 1121: '... suggesting that the effect of the Gal4 TAs on its transcription and the [the] growth rate ...'

Thanks for the catch – fixed!

Reviewer #3 (Remarks to the Author):

Thank you for addressing my comments.

Thanks for your input!

References

1. Borchardt, E. K. *et al.* Controlling mRNA stability and translation with the CRISPR endoribonuclease Csy4. *RNA* **21**, 1921–1930 (2015).
2. Salomon, W. E., Jolly, S. M., Moore, M. J., Zamore, P. D. & Serebrov, V. Single-Molecule Imaging Reveals that Argonaute Reshapes the Binding Properties of Its Nucleic Acid Guides. *Cell* **162**, 84–95 (2015).
3. Meyer, K. D. *et al.* 5' UTR m6A Promotes Cap-Independent Translation. *Cell* **163**, 999–1010 (2015).